# Roles of SmeYZ, SbiAB, and SmeDEF Efflux Systems in Iron Homeostasis of *Stenotrophomonas maltophilia*

Chao-Jung Wu,[a,b] Yu Chen,[a] Li-Hua Li,[b,c] Cheng-Mu Wu,[a] Yi-Tsung Lin,[d,e] Cheng-Hua Ma,[f] Tsuey-Ching Yang[a]

[a]Department of Biotechnology and Laboratory Science in Medicine, National Yang Ming Chiao Tung University, Taipei, Taiwan
[b]School of Medical Laboratory Science and Biotechnology, College of Medical Science and Technology, Taipei Medical University, Taipei, Taiwan
[c]Department of Pathology and Laboratory Medicine, Taipei Veterans General Hospital, Taipei, Taiwan
[d]Division of Infectious Diseases, Department of Medicine, Taipei Veterans General Hospital, Taipei, Taiwan
[e]School of Medicine, National Yang Ming Chiao Tung University, Taipei, Taiwan
[f]Department of Chemistry, National Taiwan University, Taipei, Taiwan

**ABSTRACT** *Stenotrophomonas maltophilia*, a nonfermenting Gram-negative rod, is frequently isolated from the environment and is emerging as a multidrug-resistant global opportunistic pathogen. *S. maltophilia* harbors eight RND-type efflux pumps that contribute to multidrug resistance and physiological functions. Among the eight efflux pumps, SmeYZ pump is constitutively highly expressed. In our previous study, we demonstrated that loss-of-function of the SmeYZ pump results in pleiotropic phenotypes, including abolished swimming motility, decreased secreted protease activity, and compromised tolerance to oxidative stress and antibiotics. In this study, we attempted to elucidate the underlying mechanisms responsible for Δ*smeYZ*-mediated pleiotropic phenotypes. RNA-seq transcriptome analysis and subsequent confirmation with qRT-PCR revealed that *smeYZ* mutant experienced an iron starvation response because the genes involved in the synthesis and uptake of stenobactin, the sole siderophore of *S. maltophilia*, were significantly upregulated. We further verified that *smeYZ* mutant had low intracellular iron levels via inductively coupled plasma mass spectrometry (ICP-MS). Also, KJΔYZ was more sensitive to 2,2′-dipyridyl (DIP), a ferrous iron chelator, in comparison with the wild type. The contribution of SmeYZ, SmeDEF, and SbiAB pumps to stenobactin secretion was suggested by qRT-PCR and further verified by Chrome Azurol S (CAS) activity, iron source utilization, and cell viability assays. We also demonstrated that loss-of-function of SmeYZ led to the compensatory upregulation of SbiAB and SmeDEF pumps for stenobactin secretion. The overexpression of the SbiAB pump resulted in a reduction in intracellular iron levels, which may be the key factor responsible for the Δ*smeYZ*-mediated pleiotropic phenotypes, except for antibiotic extrusion.

**IMPORTANCE** Efflux pumps display high efficiency of drug extrusion, which underlies their roles in multidrug resistance. In addition, efflux pumps have physiological functions, and their expression is tightly regulated by various environmental and physiological signals. Functional redundancy of efflux pumps is commonly observed, and mutual regulation occurs among these functionally redundant pumps in a bacterium. *Stenotrophomonas maltophilia* is an opportunistic pathogen that shows intrinsic multi-drug resistance. In this study, we demonstrated that SmeYZ, SbiAB, and SmeDEF efflux pumps of *S. maltophilia* display functional redundancy in siderophore secretion. Inactivation of *smeYZ* led to the upregulation of *smeDEF* and *sbiAB*. Unexpectedly, *sbiAB* overexpression resulted in the reduction of intracellular iron levels, which led to pleiotropic defects in *smeYZ* mutant. This study demonstrates a previously unidentified connection between efflux pumps, siderophore secretion, and intracellular iron levels in *S. maltophilia*.

**KEYWORDS** efflux pump, iron homeostasis, *Stenotrophomonas maltophilia*

Address correspondence to Tsuey-Ching Yang, tcyang@nycu.edu.tw.

The authors declare no conflict of interest.

Iron acts as a universal redox catalyst and participates in numerous biological processes; thus, it is an essential trace metal required by almost all living organisms, including pathogens and their hosts. It is not surprising that violent competition for iron typically occurs between the host and pathogen in the infection niches (1). Host utilizes nutritional immunity to sequester iron from pathogens. To overcome nutritional immunity, bacteria employ two major methods to acquire iron. First, bacteria can directly acquire iron from hosts (such as heme and transferrin), environment (such as ferric citrate and ferrous iron), or other bacteria inhabiting the same niches (xenosiderophore) in a receptor-mediated manner. Second, bacteria can synthesize and secrete diverse iron chelators, such as siderophores, hemophores, and citrate, into the environment for seizing iron. Siderophores are among the strongest known binders of Fe(III). A bacterium generally harbors at least one endogenous siderophore synthesis system for iron acquisition such as enterobactin of *Escherichia coli* and pyoverdine and pyochelin of *Pseudomonas aeruginosa*. Successful siderophore-mediated iron uptake must include siderophore biosynthesis, siderophore secretion, and ferri-siderophore uptake and utilization (2).

Siderophores are potent iron chelators. The newly synthesized siderophores may also be toxic to the cells that produce them because of the inadvertent chelation of essential iron from internal biomolecules (2). Thus, an efficient export mechanism needs to be coupled with siderophore biosynthesis. In bacteria, the siderophore secretion gene and siderophore synthesis genes are commonly organized as an operon or a gene cluster (3). In *E. coli*, a total of 16 enterobactin-associated genes are clustered together, including enterobactin synthesis genes (*entA*, *entB*, *entC*, *entD*, *entE*, *entF*, *entH*, and *ybdZ*), enterobactin secretion gene (*entS*), enterobactin uptake genes (*fepA*, *fepB*, *fepC*, *fepD*, *fepE*, and *fepG*), and enterobactin hydrolysis gene (*fes*) (3).

The efflux pump, a transmembrane transporter, is a highly conserved system in most microorganisms to expel various toxic compounds from cells (4). The resistance-nodulation-division (RND)-type efflux pump comprises an inner membrane transporter, a periplasmic membrane fusion protein, and an outer membrane protein, forming a transmembrane tripartite complex. Clinically significant multidrug resistance (MDR) is known to be caused by the overexpression of RND-type efflux pumps (5). The involvement of RND-type pumps in physiological functions is also reported, including cell-to-cell communication, siderophore secretion, stress alleviation (oxidative, nitrosative, envelope, and acid stress), biofilm formation, and virulence (4).

*Stenotrophomonas maltophilia* is generally an environmental organism but is also an opportunistic pathogen that can infect a broad range of hosts (6). The released genome sequences and reported studies have revealed that *S. maltophilia* produces a single catecholate-type siderophore, stenobactin, in response to iron depletion stress (7). The proteins encoded by the *entSCEBB'FA* operon participate in the synthesis of stenobactin (8). In addition, FepA (Smlt1426) is known to be a specific receptor for the uptake of ferri-stenobactin (7). In contrast to stenobactin synthesis and uptake systems, the stenobactin secretion system of *S. maltophilia* is poorly understood.

*S. maltophilia* has gained much attention owing to its resistance to a wide range of antibiotics (9). The involvement of tripartite efflux pumps of SmeABC, SmeDEF, SmeGH, SmeIJK, SmeOP, SmeVWX, SmeYZ, MacABCsm, and EmrCAB in antibiotic resistance has been revealed (10–18). In addition to antibiotic resistance, the significance of some efflux pumps in physiological functions has also been highlighted. For example, SmeVWX and MacABCsm alleviate oxidative stress, and SmeIJK alleviates envelope stress (13, 17, 19). SmeYZ is an intrinsically expressed pump, like the housekeeping pumps MexAB-OprM in *P. aeruginosa* and AcrAB-TolC in *E. coli* (4), supporting its important role in essential bacterial physiology. In our previous study, we have revealed that loss-of-function of SmeYZ significantly compromised swimming motility, secreted protease activity, and tolerance toward oxidative stress and some antibiotics (16). We aimed to elucidate the underlying mechanism responsible for Δ*smeYZ*-mediated pleiotropic defects in this study.

## RESULTS

**Iron starvation response occurs in *smeYZ* mutant.** To elucidate the mechanism underlying the pleiotropic phenotypes of KJΔYZ, a *smeYZ* isogenic deletion mutant of *S. maltophilia* KJ (16), RNA-seq transcriptome analysis was conducted to compare the transcriptional profiles of wild-type KJ and KJΔYZ. A relative change in transcript level (transcripts per kilobase million, TPM) equal to or greater than 3-fold was defined as differentially expressed. Among the 4,475 genes assayed, 434 genes were differentially expressed in KJΔYZ, including 218 (4.9%) upregulated genes and 216 (4.8%) downregulated genes (Table S1, NCBI SRA accession number PRJNA801053.). Gene enrichment analysis of the 218 upregulated genes indicated a significant enrichment of genes involved in iron homeostasis, amino acid (valine, isoleucine, asparagine, and histidine) biosynthesis, and flagellum-mediated motility (Fisher's exact test, $P < 0.01$) (Fig. 1A). Of note, genes predicted or known to be involved in the stenobactin synthesis and different iron source uptake systems were enriched (Table 1). This observation suggested that the inactivation of *smeYZ* causes a disturbance in cellular iron homeostasis, resembling an iron starvation condition. Thus, we further carried out the transcriptome analysis of KJ cells without and with the treatment of 2,2′-dipyridyl (DIP) for comparison. DIP is a ferrous iron chelator that can reduce intracellular iron pools and generate iron depletion stress. Transcriptome analysis revealed that 229 and 230 genes were up- and downregulated, respectively, in DIP-treated KJ cells. A similar strategy was used for the gene enrichment analysis, and the results are shown in Fig. 1A and Fig. S1.

To examine the transcriptome correlation between KJΔYZ and DIP-treated KJ, Pearson correlation coefficient was calculated. The Pearson coefficient was observed to be 0.88, suggesting a high correlation between the transcriptome of KJΔYZ and DIP-treated KJ cells. Furthermore, we also noticed that 66 and 58 genes were upregulated and downregulated, respectively, in both KJΔYZ and DIP-treated KJ cells (Fig. S2).

We tested the expression of selected genes in KJ, KJΔYZ, and DIP-treated KJ via qRT-PCR to verify the transcriptome results. The genes assayed included *entE* (Smlt2821) (7); *fepA* (Smlt1426) (7); *pacA* (Smlt2666) (20); Smlt0795, a homolog of hemin uptake receptor *hmuR* (21); and Smlt2858, a homolog of ferric citrate receptor *fecA* (22). Compared with wild-type KJ, these genes were upregulated in KJΔYZ- and DIP-treated KJ (Fig. S3), thereby validating the transcriptome results.

To further confirm whether KJΔYZ cells were under iron starvation conditions, the cell viabilities of KJ and KJΔYZ in LB agar with or without DIP were evaluated. Consistent with our previous reports, the growth of wild-type KJ was hardly affected by 20 μg/mL DIP, moderately compromised by 30 μg/mL DIP, and almost abolished by 40 μg/mL DIP (8). Nevertheless, the viability of KJΔYZ was apparently affected by 20 μg/mL DIP and was mostly inhibited by 30 μg/mL DIP (Fig. 1B). Given that KJΔYZ displayed an observed growth compromise in DIP-null LB agar compared with wild-type KJ, we further checked the DIP tolerance of strains KJ and KJΔYZ by CFU counting and consistent results were obtained (Fig. S4). Collectively, relative to wild-type KJ, KJΔYZ had poor tolerance to iron depletion stress. Next, the intracellular iron concentrations in wild-type KJ and KJΔYZ were determined using inductively coupled plasma mass spectrometry (ICP-MS). The iron concentration in KJΔYZ was significantly lower than that in wild-type KJ (Fig. 1C).

We attempted to complement KJΔYZ with a *smeYZ*-containing plasmid by conjugation. However, the conjugation efficiency was extremely low for KJΔYZ, and no stable transconjugant KJΔYZ(pSmeYZ) was obtained. Hence, the complementation strain KJΔYZ(pSmeYZ) was not included in the following experiments.

**Blocking stenobactin synthesis in KJΔYZ restores the intracellular iron levels and rescues pleiotropic defects.** The above results suggested that stenobactin synthesis was upregulated in KJΔYZ (Table 1; Fig. S3); however, KJΔYZ still experienced iron depletion (Fig. 1B, C). The involvement of RND-type efflux pumps in siderophore secretion has been reported (23). Hence, we hypothesized that deletion of *smeYZ* could compromise the secretion of stenobactin and result in its intracellular accumulation. High level of stenobactin can subsequently chelate iron from iron-rich proteins, and

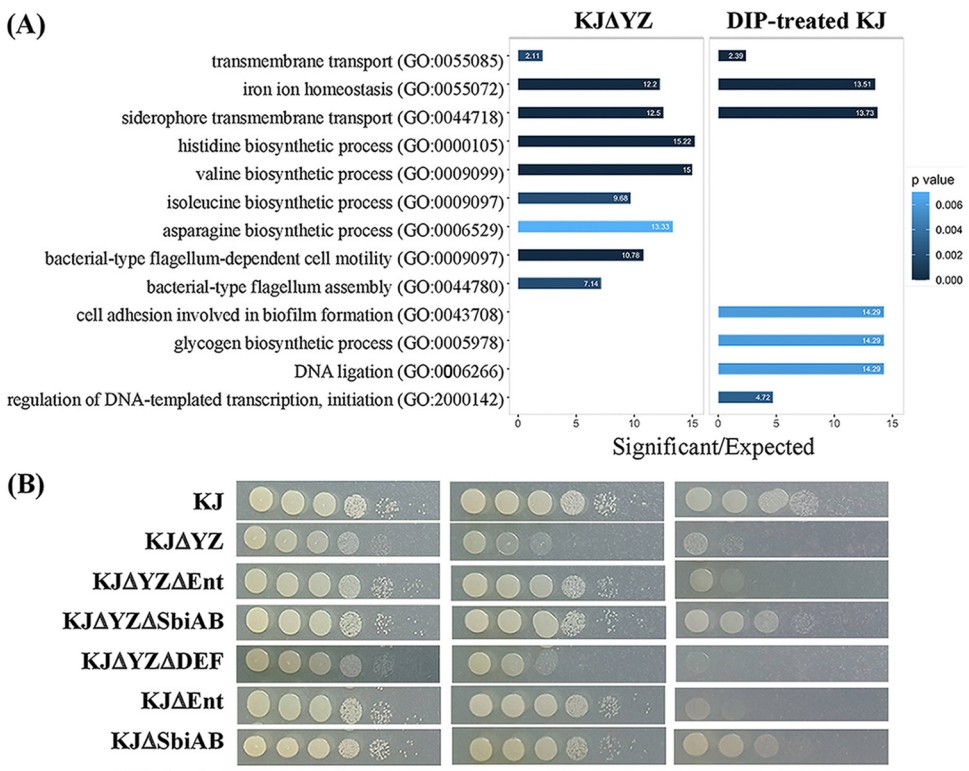

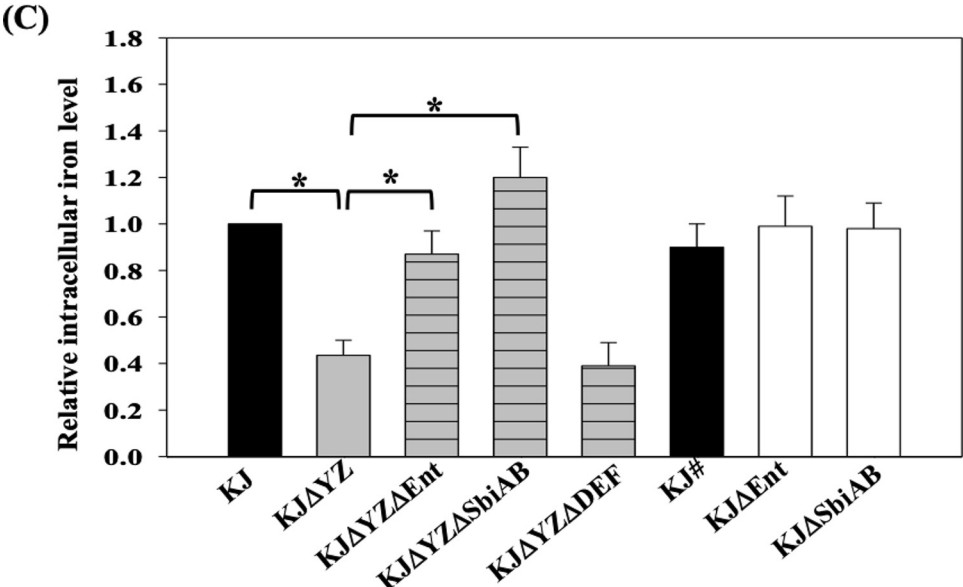

**FIG 1** *SmeYZ* mutant, KJΔYZ, exhibits an iron starvation response. (A) Gene ontology classification of upregulated genes in KJΔYZ and DIP-treated KJ. Gene enrichment analysis on the upregulated genes was performed using topGO package with Fisher exact test and weighted 01 algorithm. The GO terms with $P < 0.01$ were selected as significantly enriched functional groups. The bar showed the fold enrichment of the GO term. (B) The cell viabilities of KJ and its derived mutants in DIP-containing LB agar. Bacterial cells of $2 \times 10^5$ CFU/$\mu$L were 10-fold serially diluted, spotted onto LB agar without or with DIP as indicated, and incubated at 37°C for 24 h. (C) The intracellular iron levels of wild-type KJ and its derived mutants. Overnight-cultured bacterial cells were inoculated into fresh LB medium and incubated for 5 h. The amounts of intracellular iron in the strains assayed were determined by inductively coupled plasma mass spectrometry (ICP-MS). The relative iron levels were calculated using the iron level of 5-h cultured KJ cells as 1. #, 18-h cultured KJ cells. Data are the means from three independent experiments. Error bars represent the standard deviation for triplicates. *, $P < 0.05$, significance calculated by Student's *t* test.

**TABLE 1** The differentially expressed genes, selected from transcriptome enrichment analysis, of KJΔYZ and DIP-treated KJ compared with wild-type KJ

| Locus Smlt | TPM[b] | | Fold change[c] | TPM | Fold change | Encoded protein |
|---|---|---|---|---|---|---|
| | KJ | KJΔYZ | KJ vs KJΔYZ | KJ + DIP | None vs DIP-treated | |
| 0009 | 606.55 | 1540.12 | +2.54 | 2934.91 | +4.84 | TonB, energy transducer |
| 0487 | 71.14 | 378.23 | +5.32 | 567.66 | +7.98 | ygjH/viuB, ferric reductase |
| 0488 | 41.39 | 211.17 | +5.10 | 367.92 | +8.89 | PadR family transcriptional regulator |
| 0602 | 11.40 | 40.65 | +3.57 | 2.51 | −4.53 | TonB dependent receptor protein |
| 0603 | 2.48 | 20.76 | +8.39 | 1.40 | −1.77 | carboxypeptidase A |
| 0794 | 10.05 | 52.90 | +5.26 | 3312.74 | +329.53 | HemP, heme uptake protein, |
| 0795 | 2.51 | 86.44 | +34.49 | 1420.40 | +566.70 | HemA, TonB-dependent outer membrane receptor |
| 0796 | 1.13 | 47.00 | +41.56 | 880.11 | +778.21 | HemB |
| 0797 | 4.47 | 100.07 | +22.37 | 1050.66 | +234.87 | HemC |
| 1144c | 2.71 | 15.77 | +5.82 | 5.65 | +2.08 | TonB-dependent outer membrane receptor |
| 1148 | 7.58 | 42.84 | +5.65 | 488.70 | +64.45 | TonB-dependent outer membrane receptor |
| 1233 | 2.58 | 31.79 | +12.33 | 296.30 | +114.91 | TonB-dependent outer membrane receptor |
| **1426**[a] | 23.29 | 146.79 | +6.30 | 2791.57 | +119.87 | FepA, TonB-dependent outer membrane receptor |
| 1428 | 119.90 | 3719.86 | +31.03 | 354.35 | +2.96 | cation efflux protein |
| 1762 | 3.08 | 27.32 | +8.88 | 263.39 | +85.59 | TonB-dependent outer membrane receptor |
| 2353 | 15.09 | 45.28 | +3.00 | 1107.43 | +73.37 | alpha/beta hydrolase |
| 2354 | 4.87 | 179.25 | +36.78 | 2000.26 | +410.47 | ATP-binding protein |
| 2355 | 4.16 | 201.80 | +48.46 | 1813.75 | +435.50 | ABC transporter substrate-binding protein |
| 2356 | 1.08 | 58.81 | +54.29 | 330.69 | +305.27 | iron ABC transporter permease |
| 2357 | 2.18 | 64.20 | +29.49 | 610.07 | +280.26 | ABC transporter ATP-binding protein |
| 2642 | 13.19 | 57.86 | +4.39 | 12.49 | −1.06 | SbiA |
| 2643 | 13.91 | 73.90 | +5.31 | 13.24 | −1.05 | SbiB |
| 2664 | 4.74 | 16.96 | +3.58 | 120.75 | +25.48 | SpaI, FecI-like sigma factor |
| 2665 | 4.87 | 20.90 | +4.29 | 204.34 | +41.98 | SpaR, FecR-like TonB-dependent receptor |
| 2666 | 2.51 | 22.94 | +9.13 | 114.16 | +45.43 | SpaA, FecA-like receptor |
| 2712 | 2.59 | 62.09 | +23.97 | 802.47 | +309.72 | outer membrane protein |
| 2713 | 4.83 | 525.04 | +108.72 | 9564.36 | +1980.49 | extracellular protein |
| 2714 | 5.38 | 254.29 | +47.23 | 3390.63 | +629.8 | FecA-like TonB-dependent receptor |
| 2715 | 11.34 | 59.90 | +5.28 | 310.74 | +27.39 | FecR-like protein |
| 2716 | 17.05 | 25.09 | +1.47 | 322.90 | +18.93 | FecI-like RNA polymerase sigma factor |
| **2817** | 5.36 | 63.24 | +11.79 | 575.72 | +107.33 | EntA, 2,3-dihydro-2,3-dihydroxybenzoate dehydrogenase |
| **2818** | 4.19 | 91.64 | +21.89 | 1177.56 | +281.35 | EntF, enterobactin synthase |
| **2819** | 0.00 | 0.00 | 0.00 | 0.00 | 0.00 | EntB′, bifunctional isochorismatase/aryl carrier protein |
| **2820** | 4.17 | 99.36 | +23.83 | 1952.19 | +468.28 | EntB, bifunctional isochorismatase/aryl carrier protein |
| **2821** | 2.10 | 77.81 | +37.08 | 1520.03 | +724.47 | EntE, (2,3-dihydroxybenzoyl)adenylate synthase |
| **2822** | 0.63 | 21.04 | +33.32 | 409.80 | +648.97 | EntC, isochorismate synthase |
| 2823 | 3.59 | 22.80 | +6.35 | 607.56 | +169.22 | EntS, MFS transporter |
| 2858 | 3.37 | 18.78 | +5.58 | 145.31 | +43.15 | TonB-dependent outer membrane receptor |
| 2935 | 0.89 | 3.91 | +4.41 | 219.80 | +247.80 | FecI-like RNA polymerase sigma factor |
| 2936 | 1.87 | 5.56 | +2.97 | 384.21 | +205.39 | FecR-like protein |
| 2937 | 0.91 | 7.47 | +8.17 | 311.95 | +340.93 | FecA-like TonB-dependent receptor |
| 2938 | 1.32 | 12.77 | +9.69 | 383.34 | +290.9 | iron regulated lipoprotein |
| 2939 | 0.84 | 8.27 | +9.79 | 387.97 | +459.26 | TonB, energy transducer |
| 3022 | 7.46 | 35.46 | +4.75 | 629.93 | +84.42 | TonB-dependent outer membrane receptor |
| 3094 | 78.42 | 504.68 | +6.44 | 60.58 | −1.29 | TonB, energy transducer |
| 3477 | 111.30 | 153.35 | +1.38 | 47.18 | −2.36 | TonB, energy transducer |
| 3645 | 248.72 | 1716.43 | +6.90 | 104.32 | −2.38 | TonB-dependent outer membrane receptor |
| 3789 | 31.01 | 146.11 | +4.71 | 89.63 | +2.89 | TonB-dependent outer membrane receptor |
| 3892 | 1.99 | 11.90 | +5.98 | 34.57 | +17.37 | TonB, energy transducer |
| 3893 | 7.43 | 20.31 | +2.73 | 77.33 | +10.4 | ExbD, TonB system transport protein |
| 3894 | 6.52 | 24.08 | +3.70 | 95.07 | +14.59 | ExbB, TonB-system energizer |
| 3896 | 5.51 | 19.38 | +3.52 | 67.21 | +12.19 | heme oxygenase |
| 3898 | 2.22 | 14.91 | +6.71 | 92.66 | +41.71 | FecA-like TonB-dependent receptor |
| 3899 | 9.67 | 13.69 | +1.42 | 56.21 | +5.82 | FecR-like protein |
| 3900 | 9.03 | 19.59 | +2.17 | 630.59 | +69.84 | FecI-like RNA polymerase sigma factor |
| 3999 | 3.07 | 47.07 | +15.32 | 1314.56 | +427.96 | TonB-dependent outer membrane receptor |
| 4003 | 6.32 | 30.26 | +4.79 | 5.64 | −1.12 | TonB-dependent outer membrane receptor |
| 4026 | 21.96 | 129.57 | +5.90 | 21.91 | −1.00 | TonB-dependent outer membrane receptor |
| 4070 | 27.18 | 129.33 | +4.76 | 34.97 | +1.29 | SmeF |
| 4071 | 36.31 | 226.19 | +6.23 | 41.95 | +1.16 | SmeE |

**TABLE 1** (Continued)

| Locus Smlt | TPM[b] | | Fold change[c] | TPM | Fold change | Encoded protein |
| | KJ | KJΔYZ | KJ vs KJΔYZ | KJ + DIP | None vs DIP-treated | |
| --- | --- | --- | --- | --- | --- | --- |
| 4072 | 36.14 | 228.24 | +6.32 | 48.73 | +1.35 | SmeD |
| 4135 | 16.18 | 58.43 | +3.61 | 1780.96 | +110.09 | TonB-dependent outer membrane receptor |

[a]Bold letters, the genes are known to participate in ferri-stenobactin uptake (Smlt1426) and stenobactin synthesis (Smlt2822 to Smlt2817).

[b]TPM, transcripts per kilobase million.

[c]Negative fold changes represent genes that were significantly downregulated in response to *smeYZ* inactivation or DIP treatment, whereas positive fold changes represent upregulation in response to *smeYZ* deletion or DIP treatment.

thus, damage protein functions, thereby causing pleiotropic defects. If this is true, we should be able to specifically rescue the dysfunctions in KJΔYZ by blocking stenobactin synthesis in KJΔYZ. KJΔEnt, a stenobactin-null mutant, was constructed in our previous study (20). We introduced a Δ*smeYZ* allele into the KJΔEnt chromosome, generating KJΔYZΔEnt. Relative to KJΔYZ, KJΔYZΔEnt had restored several functions to the wild-type levels, including DIP tolerance, intracellular iron level, swimming motility, oxidative stress tolerance, and secreted protease activity (Fig. 1B, C, and 2A to D), except susceptibility to aminoglycosides (AGs). Like KJΔYZ, KJΔYZΔEnt was more susceptible to tobramycin than KJΔEnt (Fig. 2E), demonstrating that the positive effect of *smeYZ* deletion on AG susceptibility was unrelated to stenobactin. Complementation of KJΔYZΔEnt with plasmid pSmeYZ restored AG resistance (Fig. 2E). These results supported that the pleiotropic defects, except AG susceptibility, in KJΔYZ need to be ascribed to Δ*smeYZ*-mediated stenobactin synthesis. In contrast, Δ*smeYZ*-mediated increment in the AG susceptibility highly attributed to the extrusion ability of the SmeYZ pump.

**SmeYZ, SmeDEF, and SbiAB contribute to stenobactin secretion.** Based on the above results, we hypothesized that the SmeYZ pump participates in stenobactin secretion. To test this notion, Chrome Azurol S (CAS) activity and iron source utilization assays were performed. An iron source utilization assay was used to investigate the ability of stenobactin in cell-free culture supernatant to support the growth of iron-starved KJΔEnt. In our previously established protocol, stenobactin synthesis and secretion were achieved upon 20 μg/mL DIP treatment in wild-type KJ (8). Considering that KJΔYZ exhibited compromised viability in 20 μg/mL DIP-containing medium (Fig. 1B), the DIP-treatment protocol may cause a bias. In several bacteria, siderophore synthesis and secretion are repressed by the Fur repressor (24). Thus, the inactivation of *fur* would result in the de-repression of genes associated with stenobactin synthesis and secretion. This characteristic made the *fur* mutant a suitable strain for the study of stenobactin secretion. The Δ*smeYZ* allele was introduced into the *fur* mutant, KJΔFur, to yield KJΔFurΔYZ for the following assay (8). Meanwhile, KJΔFurΔEnt, a stenobactin-null *fur* mutant, was also prepared as a negative control.

The cell-free culture supernatant collected from KJΔFur had a CAS activity of 20.1 ± 1.2 U (Fig. 3A) and supported KJΔEnt growth with the growth zones of 16.8 ± 0.5 nm (Fig. 3B); in contrast, merely 0.3 ± 0.05 U CAS activity was detected (Fig. 3A) and no growth zone was observed in the KJΔFurΔEnt counterpart (Fig. 3B), indicating that stenobactin is the key molecule responsible for CAS activity and capturing the ferric iron to support the growth of iron-starved KJΔEnt. Compared with that from KJΔFur, the cell-free culture supernatant collected from KJΔFurΔYZ displayed a lower CAS activity (16.4 ± 0.3 U) (Fig. 3A) as well as a reduced ability to support the growth of KJΔEnt, with a growth zone of 12.5 ± 1.0 nm (Fig. 3B). Nevertheless, we also noticed that cell-free culture supernatant collected from KJΔFurΔYZ still had significant CAS activity (Fig. 3A) and was able to support the growth of KJΔEnt (Fig. 3B). Collectively, these results supported the involvement of the SmeYZ pump in stenobactin secretion and suggested that other mechanisms, in addition to SmeYZ, also contribute to stenobactin secretion.

We compared the expression of 11 tripartite efflux pumps (SmeABC, SmeDEF, SmeGH, SmeIJK, SmeMN, SmeOP, SmeVWX, EmrCAB, Smlt1443-1444, MacABCsm, Smlt2642-2643) in wild-type KJ and KJΔFur to elucidate their involvement in stenobactin secretion. Among

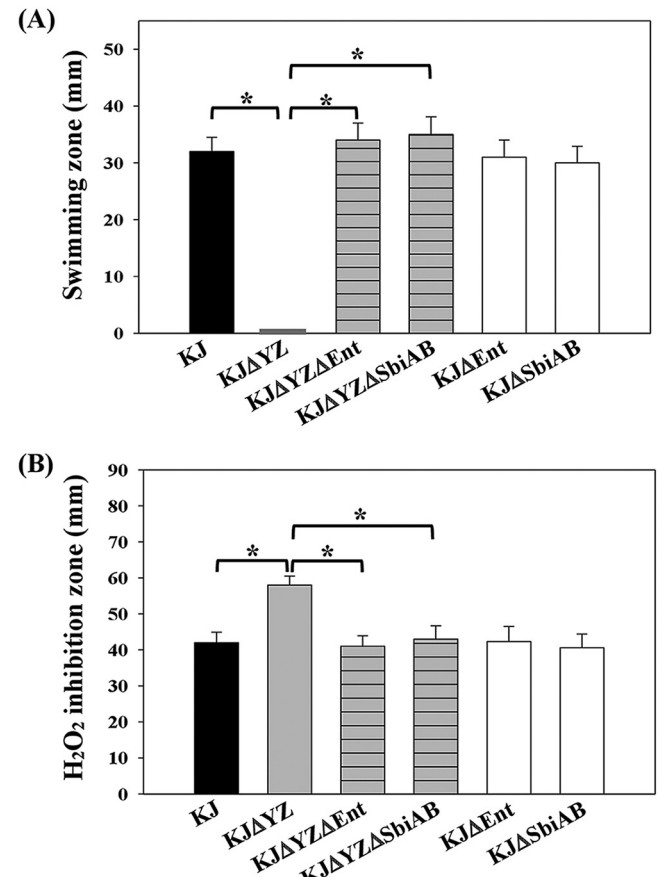

**FIG 2** Functional assays of wild-type KJ, KJΔYZ, KJΔYZΔEnt, KJΔYZΔSbiAB, KJΔEnt, and KJΔSbiAB. (A) Swimming motility assay. Five $\mu$L of bacterial cell suspension was inoculated onto the swimming agar (1% tryptone, 0.5% NaCl, and 0.15% agar). After a 48-h incubation at 37°C, the swimming zones were recorded. Data are the means from three independent experiments. Error bars indicate the standard deviations for three triplicate samples. *, $P < 0.05$, significance calculated by Student's $t$ test. (B) $H_2O_2$ susceptibility assay. Bacterial cells suspension tested were uniformly spread onto the MH agar. Sterile filter paper with 20 $\mu$L of 10% $H_2O_2$ was placed onto LB agar. After a 24-h incubation at 37°C, the diameter of a zone of growth inhibition was measured. Data are the means from three independent experiments. Error bars indicate the standard deviations for three triplicate samples. *, $P < 0.05$, significance calculated by Student's $t$ test. (C) Menadione tolerance assay. The logarithmic-phase bacterial cells of $2 \times 10^5$ CFU/$\mu$L were 10-fold serially diluted. Five $\mu$L of bacterial suspension were spotted onto the LB agar without or with 40 $\mu$g/mL MD as indicated. The cell viability was recorded after a 24-h incubation at 37°C. (D) Secreted protease activity assay. Forty $\mu$L of bacterial cells suspension was dipped onto LB agar containing 1% skim milk. The secreted proteolytic activity of the bacteria was assessed by measuring the transparent zones around the bacteria after a 72-h incubation at 37°C. Data are the means from three independent experiments. Error bars indicate the standard deviations for three triplicate samples. *, $P < 0.05$, significance calculated by Student's $t$ test. (E) Tobramycin susceptibility assay. The logarithmic-phase bacterial cells of $2 \times 10^5$ CFU/$\mu$L were 10-fold serially diluted. Five $\mu$L of bacterial suspension were spotted onto the MH agar without or with 50 $\mu$g/mL tobramycin as indicated. The cell viability was recorded after a 24-h incubation at 37°C.

the 11 genes tested, *smeE* and *Smlt2642* transcripts were upregulated (Fig. 3C). We surveyed whether the putative Fur binding sites (25) were found upstream the *smeD* and Smlt2643 genes, but no positive results were obtained, suggesting that Fur may indirectly regulate the expression of *smeDEF*, and *Smlt2642-2643*. Nevertheless, we noticed that the genes upstream *smeD* and Smlt2643 encode a TetR-type transcriptional regulator and a two-component regulatory system, respectively. SmeDEF, an RND-type efflux pump, is known to be involved in acquired multidrug resistance (11). The proteins encoded by *Smlt2642* and *Smlt2643* are organized into an ABC-type efflux pump. Based on our study, we designated Smlt2643 and Smlt2642 as SbiA and SbiB hereafter, with Sbi representing <u>s</u>teno<u>b</u>actin and <u>i</u>ron. To assess the contribution of SmeDEF and SbiAB to stenobactin

**(C)**

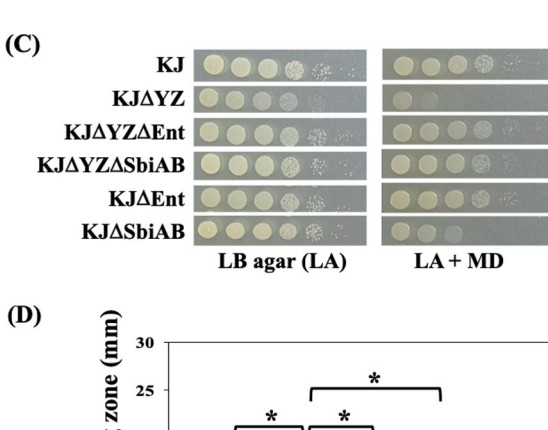

**(D)**

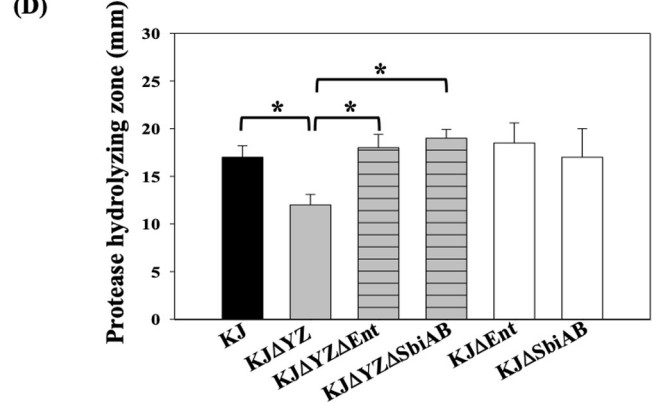

**(E)**

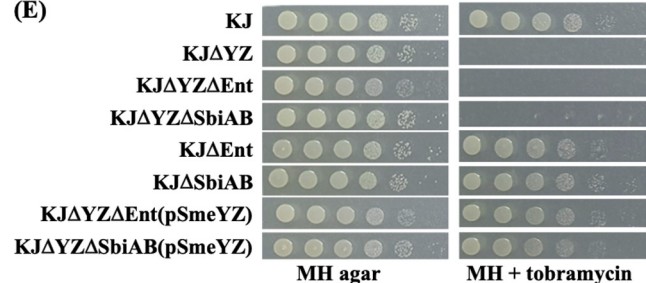

**FIG 2** (Continued)

secretion, *smeYZ*-, *sbiAB*-, and *smeDEF*-derived in-frame deletion mutants, either alone or in combination, were introduced into KJΔFur, and the cell-free culture supernatants collected from the resultant mutants were subjected to CAS activity and iron-supporting growth assays.

Fig. 3A shows the CAS activities of the KJΔFur-derived pump mutants. The CAS activity of KJΔFurΔDEF was lower than that of KJΔFur, and no significant CAS activity was detected in the cell-free culture supernatants of KJΔFurΔYZΔDEF (Fig. 3A), suggesting that SmeYZ and SmeDEF contribute to stenobactin secretion. The CAS activities of KJΔFur and KJΔFurΔSbiAB were comparable (Fig. 3A), seemingly ruling out the involvement of SbiAB in stenobactin secretion. However, when we further compared the CAS activities of KJΔFurΔYZ and KJΔFurΔYZΔSbiAB as well as KJΔFurΔDEF and KJΔFurΔDEFΔSbiAB (Fig. 3A), the mild contribution of SbiAB to stenobactin secretion was revealed. A similar tendency was also observed in the iron source utilization assay (Fig. 3B).

To further check the contribution of SmeYZ, SmeDEF, and SbiAB to stenobactin secretion, the viabilities of mutants in iron-limited FeCl$_3$-supplemented medium were evaluated. The viabilities of the tested strains in the iron-limited FeCl$_3$-supplemented medium could have been proportional to the secreted stenobactin of these strains. Compared with wild-type KJ, single mutant KJΔYZ exhibited moderately compromised viability, double mutants KJΔYZΔSbiAB and KJΔYZΔDEF exhibited severe viability defects, and the triple mutant KJΔYZΔDEFΔSbiAB completely lost viability in the iron-limited FeCl$_3$-supplemented medium (Fig. 3D). Collectively, SmeYZ, SmeDEF, and

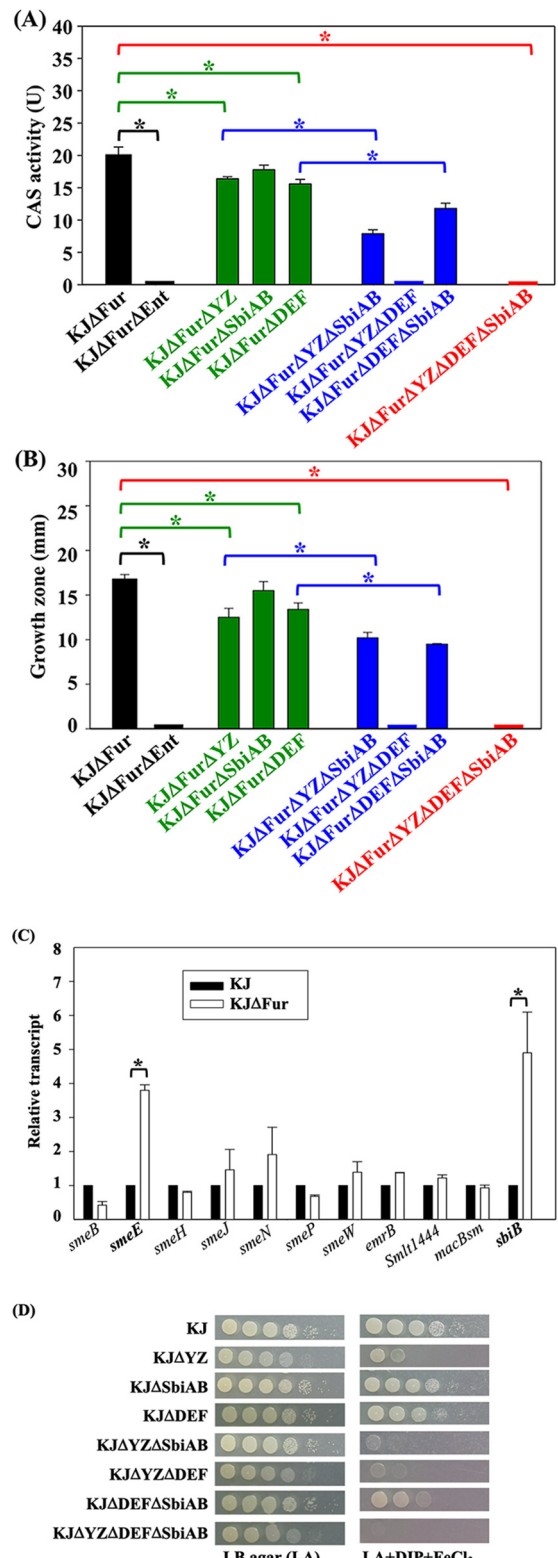

**FIG 3** SmeYZ, SmeDEF, and SbiAB are involved in stenobactin secretion. (A) CAS activity assay. The bacterial strains tested were grown in MH medium for 18 h. The cell-free culture supernatant was obtained by centrifugation and then filtered through a 0.22 $\mu$m filter. The filtrate was mixed with CAS solution and the $A_{630nm}$ was recorded. KJΔFurΔEnt was used as the control. CAS activity was calculated by the $A_{630nm}$ difference ($\Delta A_{630nm}$) between the tested strain and KJΔFurΔEnt. One unit (U) of CAS activity was defined as the amount of iron-chelating molecules that convert 1 $\mu$M ternary complex, as calculated using an extinction coefficient of 100,000$M^{-1}cm^{-1}$ for the ternary complex at

SbiAB pumps contribute to stenobactin secretion; of these, SmeYZ and SmeDEF are dominant.

**SbiAB upregulation is linked with Δ*smeYZ*-mediated intracellular low iron level.** In the preceding paragraph, we proposed a hypothesis that Δ*smeYZ*-mediated stenobactin accumulation may chelate iron from iron-rich proteins and cause the pleiotropic effects of KJΔYZ. However, two lines of evidence made us tentatively rule out the linkage of stenobactin accumulation to pleiotropic defects of KJΔYZ. (i) KJΔFurΔYZ still can export stenobactin at a significant level to support the growth of iron-starved KJΔEnt cells (Fig. 3B). Thus, we thought that the intracellular accumulated stenobactin should not be high enough to explain the severe defects of KJΔYZ. (ii) The intracellular iron level determined by ICP-MS represents the total iron, both free form (ferric iron and ferrous iron) and bound form (such as ferritin and Fe-S proteins). If the accumulated stenobactin sequestered iron from iron-binding proteins, the sequestration effect is unlikely to explain the significant drop of iron levels in KJΔYZ (Fig. 1C). Thus, we considered the possibility that intracellular low iron level, rather than stenobactin accumulation, may be linked to the pleiotropic defects of KJΔYZ.

The alteration of a single pump expression may have a downstream effect on the expression of other efflux systems (26). Thus, we imagined that a compensatory upregulation of other pumps, rather than *smeYZ* inactivation, may be the underlying factor responsible for Δ*smeYZ*-mediated intracellular low iron levels. Subsequently, we determined which efflux pump(s) are upregulated and responsible for *smeYZ*-mediated intracellular low iron levels. We reasoned that the inactivation of such an efflux pump in the Δ*smeYZ* background would revert the intracellular iron level to the wild-type level. Among the 11 tripartite pump systems examined, SmeDEF and SbiAB were significantly upregulated in KJΔYZ (Table 1), which was verified via qRT-PCR (Fig. 4).

The Δ*smeDEF* and Δ*sbiAB* were, respectively, introduced into KJΔYZ and the intracellular iron levels of KJΔYZΔDEF and KJΔYZΔSbiAB were determined by ICP-MS. The intracellular iron level of KJΔYZΔSbiAB, but not of KJΔYZΔDEF, was restored to the wild-type level (Fig. 1C), strongly suggesting that Δ*smeYZ*-mediated *sbiAB* upregulation is a critical factor responsible for the low intracellular iron levels in KJΔYZ. If this is the case, the pleiotropic defects in KJΔYZ should disappear in KJΔYZΔSbiAB. Indeed, KJΔYZΔSbiAB exhibited comparable DIP tolerance (Fig. 1B), swimming motility (Fig. 2A), $H_2O_2$ sensitivity (Fig. 2B), menadione (MD) tolerance (Fig. 2C), and secreted protease activity (Fig. 2D) to those of wild-type KJ. Collectively, SbiAB overexpression in KJΔYZ resulted in a reduction in intracellular iron levels (Fig. 1C), which subsequently resulted in pleiotropic defects (Fig. 2). So far, we have revealed a link between the functions of the SbiAB pump to stenobactin secretion and intracellular iron level homeostasis. This is the reason for the nomenclature "Sbi" (stenobactin and iron). The *smeDEF* upregulation in KJΔYZ provides a rational explanation for why KJΔYZ can export most of the synthesized stenobactin, as evidenced by the CAS activity assay (Fig. 3A) and iron-supporting growth assay (Fig. 3B).

We were interested in assessing whether SbiAB overexpression can generally lower intracellular iron levels and trigger the expression of stenobactin synthesis genes even

**FIG 3** Legend (Continued)

630 nm. Data are the means from three independent experiments. Error bars indicate the standard deviations for three triplicate samples. *, $P < 0.05$, significance calculated by Student's *t* test. (B) Iron source utilization assay. KJΔEnt cells of $2 \times 10^5$ CFU/mL were evenly spread onto the LB agar plate containing 50 $\mu$g/mL DIP and 35 $\mu$M $FeCl_3$. Bacterial cells tested were cultured in LB broth for 15 h and the cell-free culture supernatant was collected. A 6 mm-disk containing 15 mL aliquots of supernatant was laid upon the surface of KJΔEnt-spread plate. The growth zones surrounding the disk were recorded after 24-h incubation at 37°C. Data are the means from three independent experiments. Error bars indicate the standard deviations for three triplicate samples. *, $P < 0.05$, significance calculated by Student's *t* test. (C) The role of Fur in the expression of tripartite efflux pumps. Overnight-cultured KJ and KJΔFur cells were inoculated into fresh LB with an initial $OD_{450nm}$ of 0.15. Cells were grown aerobically for 5 h before measuring the transcripts as indicated by qRT-PCR. All values were normalized to the transcript of KJ cells. Bars represent the average values from three independent experiments. Error bars represent the standard error of the mean. *, $P < 0.05$, significance calculated by Student's *t* test. (D) Cell viabilities of KJ and its derived deletion mutants in iron-limited and $FeCl_3$-supplemented medium. Bacterial cells of $2 \times 10^5$ CFU/mL were 10-fold serially diluted. Five $\mu$L of bacterial suspension were spotted onto the agars as indicated. The growth of bacterial cells was observed after 24-h incubation at 37°C. DIP, 50 $\mu$g/mL; $FeCl_3$, 35 $\mu$M.

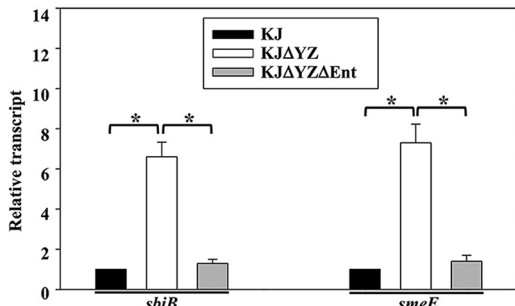

**FIG 4** *SbiAB* and *smeDEF* upregulation in KJΔYZ is stenobactin dependent. Overnight-cultured KJ, KJΔYZ, and KJΔYZΔEnt cells were inoculated into fresh LB with an initial $OD_{450nm}$ of 0.15. Cells were grown aerobically for 5 h before measuring the transcripts as indicated. All values were normalized to the transcript of KJ cells. Bars represent the average values from three independent experiments. Error bars represent the standard error of the mean. *, $P < 0.05$, significance calculated by Student's *t* test.

if *smeYZ* is functional. Thus, we determined the intracellular iron levels and *entE* transcript of KJ and KJ(pSbiAB) by ICP-MS and qRT-PCR, respectively. The intracellular iron level of KJ(pSbiAB) was slightly, but significantly, lower than that of KJ (Fig. S5A), supporting the linkage between *sbiAB* overexpression and low iron level. However, the *entE* transcripts in KJ and KJ(pSbiAB) were comparable (Fig. S5B). When the intracellular iron levels and *entE* transcript of KJΔYZ were included for comparison (Fig. S5), we assumed that the intracellular iron level of KJ(pSbiAB) is not low enough to trigger the expression of stenobactin synthesis genes.

**SbiAB and smeDEF upregulation in KJΔYZ is stenobactin dependent.** An interesting result depicted in Fig. 1C attracted our attention. The intracellular iron level of KJΔYZΔEnt was higher than that of KJΔYZ and was almost the same as that of KJ (Fig. 1C). Thus, we hypothesized that stenobactin can be the key molecule that triggers the upregulation of *smeDEF* and *sbiAB* in KJΔYZ. The *sbiB* and *smeE* transcript levels in KJ, KJΔYZ, and KJΔYZΔEnt were determined by qRT-PCR. Indeed, the *sbiB* and *smeE* transcripts showed $6.6 \pm 0.7$- and $7.3 \pm 0.9$-fold increase, respectively, in KJΔYZ, and the levels in KJΔYZΔEnt were reverted to those observed in the wild-type cells (Fig. 4).

Next, we investigated the factor(s) that trigger stenobactin synthesis in KJΔYZ. SmeYZ is constitutively highly expressed in wild-type KJ, signifying its contribution to extrude noxious metabolites generated by *S. maltophilia* grown under physiological conditions (16). The loss-of-function of SmeYZ may accumulate these noxious metabolites, which, in turn, may create a stress to trigger stenobactin synthesis. It is known that a wide variety of secondary metabolites, toxins, and noxious derivatives is synthesized during the stationary phase of bacterial growth (27). Thus, we hypothesized that stenobactin synthesis would increase during the stationary phase if these noxious metabolites are involved in triggering stenobactin synthesis. To test this notion, the expression of *entSCEBB'FA* operon expression and intracellular iron levels of logarithmic-phase and stationary-phase KJ cells were compared by $P_{entS}$-*xylE* transcriptional fusion construct and ICP-MS, respectively. The *SmeU1VWU2X* operon is intrinsically not expressed; hence, the $P_{smeU1}$-*xylE* transcriptional fusion construct, pSmeU1$_{xylE}$, was used as a negative control (15). No significant catechol 2,3-dioxygenase (C23O) activity was detected in logarithmically grown KJ cells; nevertheless, noticeable C23O activity was observed in stationary-phase KJ cells (Fig. 5). Furthermore, the intracellular iron-levels of logarithmic-phase and stationary-phase KJ cells were comparable (Fig. 1C). Collectively, these evidences support that the accumulated noxious metabolites, rather than low intracellular iron level, may be the critical stimulus to trigger the stenobactin synthesis in the stationary phase.

## DISCUSSION

Although the expression of RND-type efflux pumps is classically associated with MDR, our previous study showed that inactivation of *smeYZ*, a constitutively expressed

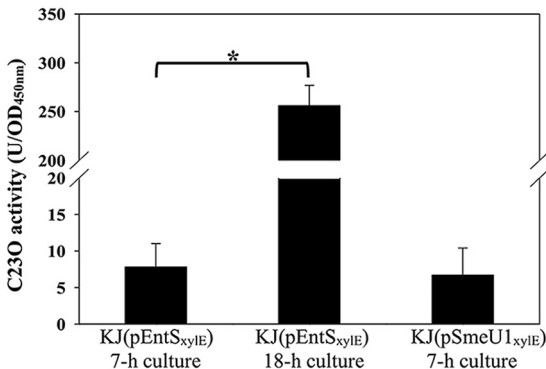

**FIG 5** Impact of bacterial growth phase on the expression of stenobactin synthesis genes. Overnight cultures of KJ(pEntSxylE) and KJ(pSmeU1xylE) were inoculated into fresh LB with an initial $OD_{450nm}$ of 0.15. Cells were grown aerobically for 7 h or 18 h before measuring the C23O activity. Bars represent the average values from three independent experiments. Error bars represent the standard error of the mean. *, $P < 0.05$, significance calculated by Student's $t$ test.

pump in *S. maltophilia*, results in manifold dysfunctions (16). Interestingly, these Δ*smeYZ*-mediated pleiotropic phenotypes are seemingly unrelated to the formally recognized efflux pump functions, such as completely abolished swimming motility. In this study, we have expanded the SmeYZ pump function for stenobactin export and demonstrated that Δ*smeYZ*-mediated pleiotropic phenotypes are attributed to the intracellular low iron levels. Similar observations have been reported for the VexGH pump in *Vibrio cholerae*. A *vexGH* mutant of *V. cholerae* exhibited impaired growth under iron-limited conditions, and this compromise was not observed in the double mutant of *vexGH* and *vibF* (a gene involved in siderophore synthesis), which is similar to KJΔYZ and KJΔYZΔEnt used in this study (28). However, unlike *S. maltophilia* KJΔYZ, *V. cholerae vexGH* mutant is not iron-stressed because the expression of genes associated with iron acquisition is not significantly upregulated in *V. cholerae vexGH* mutant (28). In contrast, KJΔYZ experienced iron-depleted conditions, very likely due to the overexpression of the SbiAB pump. SbiAB overexpression resulted in the reduction of intracellular iron levels and further augmented the iron starvation condition in KJΔYZ.

To reveal the mechanism by which *smeYZ* deletion results in pleiotropic phenotypes, we employed a strategy involving transcriptome analysis to compare DEGs between KJ and KJΔYZ as well as KJ with and without DIP treatment. Three lines of evidence support KJΔYZ being under iron starvation conditions: (i) a similar gene expression profile observed in KJΔYZ and DIP-treated KJ cells (Fig. 1A); (ii) KJΔYZ displaying a compromised DIP tolerance (Fig. 1B); and (iii) the intracellular iron level of KJΔYZ being lower than that of wild-type KJ, as revealed by ICP-Mass (Fig. 1C). In contrast to upregulated genes, there seems to be less overlap of Δ*smeYZ* and DIP-treatment transcriptome profiles for downregulated genes, except the genes associated chemotaxis (Fig. S1). This observation suggests that (i) iron levels may cause an impact on bacterial motility and (ii) the genes expression profiles of KJΔYZ seem not to be totally relevant to iron depletion; thus, KJΔYZ may encounter other stresses in addition to iron limitation. In addition, the expression of TonB genes in the transcriptome analysis attracted our attention. The TonB/ExbB/ExbD system is required to energize several TonB dependent outer membrane transporters (TBDTs), which play a role in transportation of serval nutrients such as ferri-siderophores, vitamin B12, heme, metals and oligosaccharides from outside environments (29–31). A bacterium generally harbors multiple TonB genes, whose encoded proteins seem to specifically interact with subsets of TBDTs for the transport of different substrates responsible for diverse physiological activities (32–35). There are five annotated TonB genes, Smlt0009, Smlt2939, Smlt3094, Smlt3477, and Smlt3892, in *S. maltophilia* genome. Their expression changes in KJΔYZ and DIP-treated KJ were not consistent. For example, Smlt0009 was significantly upregulated in DIP-treated KJ, but not in KJΔYZ; however, Smlt3094 was significantly upregulated in KJΔYZ, rather than in DIP-treated KJ (Table 1). These observations suggest that

*smeYZ* deletion and DIP treatment appear to cause distinct impacts on bacterial physiological activities, further confirming our aforementioned opinions that KJΔYZ displayed pleiotropic defects, not limited to iron depletion.

For a bacterium to deal with iron starvation, siderophore-mediated iron acquisition is the most efficient method. Siderophores are among the strongest known binders of ferric iron. If the newly synthesized siderophore is intracellularly accumulated, it may be toxic to the cells that produce it because of the inadvertent chelation of essential iron from internal biomolecules. The molecular mechanism underlying siderophore secretion has been revealed in some Gram-negative bacteria, and a two-step secretion model has been proposed. Inner membrane transport and tripartite efflux pumps are key components of this machinery (23, 36, 37). The inner membrane transporter is responsible for the exportation of siderophores from the cytoplasm into the periplasm (36), and the tripartite efflux pumps manage the exportation of siderophores from the periplasm to the extracellular environment (23, 37). The known inner membrane transporters involved in siderophore secretion are either MFS-type or ABC-type mono-component efflux pumps. Furthermore, the location of the inner membrane transporter gene is in the vicinity or in the same operon of siderophore synthesis genes, such as EntS in *E. coli* (36), PvdE in *P. aeruginosa* (38), LbtB in *Legionella pneumophila* (39), CsbX in *Azotobacter vinelandii* (40), NorA in *Staphylococcus aureus* (41), and AlcS in *Bordetella pertussis* and *Bordetella bronchiseptica* (42). The *entSCEBB'FA* operon in *S. maltophilia* is involved in the synthesis of stenobactin. In this operon, *entS* encodes an MFS-type inner membrane transporter, displaying 24% identity and 35% similarity with *E. coli* EntS. This feature makes it rational to consider EntS as an inner membrane transporter of stenobactin; however, this needs to be elucidated further. The known tripartite efflux pumps responsible for siderophore secretion include the RND-type efflux pumps in *E. coli* (AcrAB-TolC, AcrAD-TolC, and MdtABC-TolC) (23) and *V. cholerae* (VexGH) (28), and the ABC-type efflux pump in *P. aeruginosa* (PvdRT-OmpQ) (37). In the two-step secretion model, a siderophore is captured by tripartite efflux pumps in the periplasm rather than in the cytoplasm. This is acceptable because the tripartite efflux system is capable of capturing substrates in the periplasm (43). A secretion system for stenobactin was revealed in this study. Both RND-type (SmeYZ and SmeDEF) and ABC-type (SbiAB) tripartite efflux pumps are involved in stenobactin secretion in *S. maltophilia*. Except AcrAB-TolC in *E. coli* and SmeYZ in *S. maltophilia*, these known tripartite efflux pumps involved in siderophore secretion are generally intrinsically feebly expressed, and their expression is induced by iron-depleted conditions.

An interesting observation attracted our attention. We successfully complemented KJΔYZΔEnt and KJΔYZΔSbiAB with plasmid pSmeYZ (Fig. 2), but the same complementation failed in KJΔYZ. Compared with wild-type KJ, KJΔYZ, but not KJΔYZΔEnt and KJΔYZΔSbiAB, had lower intracellular iron level (Fig. 1C). Thus, we suggest that the complementation failure of KJΔYZ may results from its low intracellular iron level, which may have a negative impact on bacterial conjugation.

Inactivation of *smeYZ* upregulated *sbiAB* and *smeDEF* (Table 1), implying that there is functional redundancy between SmeYZ, SbiAB, and SmeDEF pumps. Indeed, our results support that SmeYZ, SbiAB, and SmeDEF have a common function in stenobactin secretion (Fig. 3). In addition, we also noticed that SbiAB overexpression, rather than SmeDEF or SmeYZ overexpression, resulted in a reduction of intracellular iron levels, although the underlying mechanism is not immediately clear. Thus, SbiAB also contributes to the homeostasis of intracellular iron level, in addition to stenobactin secretion. Fig. 6 shows the proposed mechanistic model for Δ*smeYZ*-mediated pleiotropic phenotypes. Gram-negative bacteria generally harbor an intrinsically highly expressed efflux pump, which is involved in the extrusion of physiological metabolic derivatives, such as AcrAB-TolC in *E. coli* and SmeYZ in *S. maltophilia*. Inactivation of SmeYZ may lead to metabolite accumulation, which, in some way, imposes stress on bacteria and triggers stenobactin synthesis. It has been reported that siderophore synthesis can also be regulated by other stresses, such as oxidative stress, in addition to iron depletion (44–46), and that the functions of siderophores are multidimensional beyond ferric iron capture (47). SmeYZ

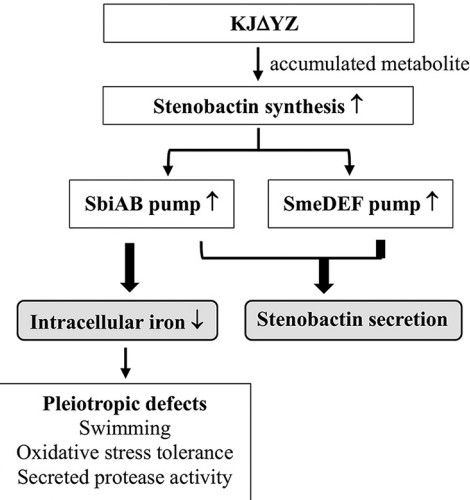

**FIG 6** Proposed mechanistic model for *smeYZ* inactivation-mediated pleiotropic defects. SmeYZ pump is an intrinsically, highly expressed efflux pump, signifying its role in the extrusion of metabolites generated from physiologically-grown KJ cells. Loss-of-function of SmeYZ may result in the accumulation of metabolites, which may trigger stenobactin synthesis. SmeYZ, SbiAB, and SmeDEF pumps contribute to stenobactin secretion; thus, inactivation of *smeYZ* compensatorily upregulates the expression of *sbiAB* and *smeDEF* to alleviate the stress of intracellular stenobactin accumulation. In addition to stenobactin secretion, SbiAB overexpression causes an impact on the reduction of intracellular iron level, which results in the pleiotropic defects of *smeYZ* mutant.

functions as an export route for stenobactin secretion; thus, inactivation of *smeYZ* can cause stress from intracellular stenobactin accumulation. In response to intracellular accumulation of stenobactin, SbiAB and SmeDEF pumps are compensatorily upregulated to manage stenobactin secretion, as SbiAB, SmeDEF, and SmeYZ have functional redundancy in stenobactin secretion. This may be the reason why the cell-free culture supernatant obtained from KJΔYZ still contained significant stenobactin for iron acquisition to support bacterial growth under iron-depleted conditions (Fig. 3B). However, Δ*smeYZ*-mediated *sbiAB* overexpression has an impact on the reduction of intracellular iron levels. The intracellular low levels of iron might be the underlying reason for the pleiotropic defects observed in *smeYZ* mutant.

## MATERIALS AND METHODS

**Bacterial strains, media, plasmids, and primers.** All bacterial strains, plasmids, and primers used in this study were listed in Table S2 and Table S3 of the supplemental material.

**Transcriptome analysis.** Overnight cultures of tested strains were subcultured into fresh LB at an initial $OD_{450}$ of 0.15 and incubated for 5 h. For KJ strain, the DIP was added to a final concentration of 30 $\mu$g/mL. Total RNA isolation, rRNA depletion, adapter-ligated cDNA library construction and enrichment, and cDNA sequencing were performed as described previously (48). The output R1 reads were mapped to the genome of KJ (GenBank accession: JAIQXD000000000) using bwa v0.7.17 (49). The gene mapping reads were counted using htseq v0.13.5 (50) with arguments stranded reverse and mode union. The total number of reads per gene between both strains was normalized by TPM values. Differential expression analysis of RNA-seq data was performed using *S. maltophilia* KJ (GenBank accession: JAIQXD000000000) as a reference. Gene enrichment analysis was conducted using topGO v2.44.0 with Fisher's exact test and Weight 01 algorithm. The Pearson correlation between KJΔYZ and DIP-treated KJ was calculated using $\log_2$ TPM values of all genes in KJ genome by cor.test function in R v4.1.1. The same protocol was used for transcriptome analysis between KJ with or without the treatment of 30 $\mu$g/mL DIP.

**Quantitative real-time-PCR.** DNA-free RNA was isolated from midexponential phase bacterial cells as described previously (15). cDNA was generated using the High Capacity cDNA Reverse transcription kit (Applied Biosystems). Quantitative reverse transcription mixture was prepared using a 1:100 dilution of the cDNA and the TaqMan Universal PCR Master Mix (Applied Biosystems). The primers used were listed in Table S3. Quantitative real-time-PCR (qRT-PCR) was carried out by the ABI Prism 7000 Sequence Detection System (Applied Biosystems). The relative fold increases in expression levels were normalized to 16s rRNA gene. Comparative quantification was carried out using the $\Delta\Delta C_T$ method (51). All experiments were run in triplicate.

**DIP, MD, and tobramycin susceptibility assay.** Overnight culture of bacterial cells was subcultured into fresh LB medium at an initial $OD_{450nm}$ of 0.15. After 5-h incubation at 37℃, bacterial culture was adjusted to an $OD_{450nm}$ of 1 and subsequently 10-fold serially diluted. Aliquots (5 $\mu$L) were spotted onto

medium with or without the supplements as indicated. The supplements were DIP of 20 and 30 $\mu$g/mL, MD of 40 $\mu$g/mL, and tobramycin of 50 $\mu$g/mL. LB agar was used for DIP and MD susceptibility assay. For tobramycin susceptibility assay, Mueller-Hinton (MH) agar was applied. The cell viabilities were recorded after 24-h incubation at 37°C. Experiments were performed at least three times and a representative result is shown.

**Inductively coupled plasma-mass spectrometry.** The bacterial culture preparation for inductively coupled plasma-mass spectrometry (ICP-MS) experiment was the same as that for the transcriptome assay. The cells were collected and washed twice with Milli-Q water and resuspended in 2 mL of Milli-Q water. Cell numbers in the cell suspension were quantified by CFU counts. Then, the cell suspension was sonicated, centrifuged, and filtered through a 0.45-$\mu$m-pore-size Millipore membrane and the aliquot was collected for ICP-MS assay. ICP-MS was performed using an Agilent 7700e instrument (Agilent Technologies, USA). The intracellular iron level was normalized by the cell numbers.

**Construction of deletion mutants.** Double cross-over homologous recombination was used to create in-frame deletion mutants. Two DNA amplicon flanking the deleted region were obtained by PCR and then subsequently ligated into the suicide vector pEX18Tc.The pEX18-derived plasmids for mutant construction were listed in Table S2. The primer sets SbiAN-F/SbiAN-R and SbiBC-F/SbiBC-R were used for the construction of plasmid p$\Delta$SbiAB. Transconjugants selection and mutant confirmation were carried out as previously described (52). Double, triple, and quadruple were constructed from the single mutant sequentially through the same procedure.

**H$_2$O$_2$ susceptibility test, swimming, and secreted protease activity assay.** The H$_2$O$_2$ susceptibility test, swimming, and secreted protease activity assay, were determined following the established protocols (16). Each experiment was performed with at least three replicates.

**Chrome Azurol S assay.** CAS assay was used to quantify the siderophore secreted in the cell-free culture supernatant as described previously with some modified (8). Overnight-cultured bacterial cells were subcultured into LB broth at an initial OD$_{450}$ of 0.15 and then cultured for 15 h at 37°C. Culture supernatants were passed through a 0.22-$\mu$m filter to obtain a cell-free supernatant for CAS assays. The filtrate was mixed with the CAS solution at a ratio of 2:1. After 30-min incubation at room temperature, the OD$_{630nm}$ was detected. One unit (U) of CAS activity was defined as the amounts of iron-chelating molecules that can convert 1 $\mu$M ternary complex, as calculated using an extinction coefficient of 100,000 M$^{-1}$cm$^{-1}$ for the ternary complex at 630 nm (53). All assays were performed at least three times.

**Iron source utilization assay.** KJ$\Delta$Ent was used as an indicator strain for iron source utilization assay since it cannot grow in an iron-depleted medium unless the exogenous iron source was supplied. Logarithmic-phase KJ$\Delta$Ent cells of $2 \times 10^5$ CFU/mL were evenly spread onto LB agar supplemented with 50 $\mu$g/mL 2,2'dipyridyl (DIP) and 35 $\mu$M FeCl$_3$. Overnight culture of strains tested was inoculated into a fresh LB broth at an initial OD$_{450nm}$ of 0.15 and then cultured for 15 h. The culture was centrifuged and the supernatant was passed through a 0.22-$\mu$m filter to get a siderophore-containing filtrate. A 6-mm disk immersed with 15 $\mu$L filtrate was applied onto the surface of KJ$\Delta$Ent-spread LB agar. The growth zones were measured after 24-h incubation at 37°C.

**Construction of promoter–*xylE* transcriptional fusion plasmids, pEntS$_{xylE}$.** The promoter region of *entSCEBB'FA* operon was obtained by PCR using the primer sets EntSN-F and EntSN-R. Plasmid pEntSxylE was constructed by cloning the PCR amplicon in front of the *xylE* gene in pRKXylE (13).

**Determination of C23O activity.** The C23O encoded by *xylE* gene can convert the colorless substrate catechol to an intensely yellow hydroxymuconic semialdehyde, which can be spectrophotometrically quantified at 375 nm. The C23O activity in intact cells was measured as described previously (48). One unit of enzyme activity (Uc) was defined as the amount of enzyme that converts 1 nmol of catechol per minute. Specific activity (Uc/OD$_{450}$) of the enzyme was defined in terms of units per OD$_{450}$ unit of cells.

**Data availability.** The whole genome shotgun sequences of *S. maltophilia* KJ has been registered under GenBank accession number JAIQXD000000000. The RNAseq data of KJ, KJ$\Delta$YZ, and DIP-treated KJ has been registered under NCBI SRA accession number PRJNA801053.

## SUPPLEMENTAL MATERIAL

Supplemental material is available online only.

**SUPPLEMENTAL FILE 1**, PDF file, 2.2 MB.

## ACKNOWLEDGMENTS

This work was supported by the Ministry of Science and Technology of Taiwan (grant number MOST 108–2320-B-010–032-MY3). The funders had no role in study design, data collection and interpretation, or the decision to submit the work for publication. The authors acknowledge the mass spectrometry technical research services from NTU Consortia of Key Technologies.

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
