## [Reviewer comments · Microbiology Spectrum]

Microbiology Spectrum

Roles of SmeYZ, SbiAB, and SmeDEF efflux systems in iron homeostasis of *Stenotrophomonas maltophilia*.

Chao-Jung Wu, Yu Chen, Li-Hua Li, Cheng-Mu Wu, Yi-Tsung Lin, Cheng-Hua Ma, and Tsuey-Ching Yang

Corresponding Author(s): Tsuey-Ching Yang, National Yang Ming Chiao Tung University

Review Timeline:

Submission Date:	December 10, 2021
Editorial Decision:	December 29, 2021
Revision Received:	February 14, 2022
Editorial Decision:	March 6, 2022
Revision Received:	March 30, 2022
Editorial Decision:	April 17, 2022
Revision Received:	April 17, 2022
Accepted:	April 30, 2022

Editor: Ayush Kumar

Reviewer(s): The reviewers have opted to remain anonymous.

Transaction Report:

DOI: <https://doi.org/10.1128/spectrum.02448-21>

December 29, 2021

Prof. Tsuey-Ching Yang
National Yang Ming Chiao Tung University
155 Section 2, Lie-Nong Street,
Taipei, Taiwan 112
Taiwan

Re: Spectrum02448-21 (Roles of SmeYZ, SbiAB, and SmeDEF efflux systems in iron homeostasis of *Stenotrophomonas maltophilia*.)

Dear Prof. Tsuey-Ching Yang:

Thank you for submitting your manuscript to Microbiology Spectrum. Your manuscript has now been reviewed by two experts in the field and they both see the work presented as valuable. However, both reviewers have suggested a number of modifications and I would like to you address these comments, before I can make a final decision on the manuscript. One reviewer also pointed out that the manuscript in its current form is also very long and could be streamlined.

Link Not Available

Sincerely,

Ayush Kumar

Journals Department
Reviewer comments:

Reviewer #1 (Public repository details (Required)):

The RNAseq data should be submitted in an appropriate public database. The authors have not provided any accession number for that in the manuscript.

Reviewer #1 (Comments for the Author):

In the manuscript Spectrum02448-2, Wu et al. reported the functions of efflux pump SmeYZ of *Stenotrophomonas maltophilia* in the export of siderophore stenobactin. I found the manuscript to be very well-written. The rationale of the experiments and the analysis of the results were clearly presented. The work is interesting and will definitely contribute to the growing body of

literature on the functional roles of efflux pumps in siderophore metabolism.

There is just one thing about this work which is unclear to me:

Lines 295, 301-303, 348-349, 365-366, 452 (Figures 1C, 3C, and 4A): Here the authors show that deletion of *smeYZ* causes low iron levels. This deletion of *smeYZ* causes upregulation of *sbiAB* and *smeDEF* which also significantly contribute to the secretion of stenobactin. How does that lead to reduction in intracellular iron levels? Wouldn't it be the opposite because upregulation of redundant efflux pumps such as *sbiAB* and *smeDEF* should reduce the toxicity of stenobactin? This is further supported by the fact that *ent* deletion restores the iron levels in *smeYZ* deletion strain. If the authors perform an experiment where *sbiAB* and *smeDEF* are overexpressed in a *smeYZ* deletion strain, shouldn't that restore iron levels? This is a major gap in their rationale in the mechanism of pleiotropy and thus the physiological role of *smeYZ*. The authors do not provide any clear rationale as to why iron is low in *smeYZ* strain.

Is the following scenario/hypothesis possible? A deletion of *smeYZ* causes stenobactin accumulation inside the cells (as the authors clearly show) causing stenobactin toxicity via iron-starvation to which the cells respond by upregulating *sbiAB* and *smeDEF* (also reported by authors). But this upregulation is not enough to release all the accumulated intracellular stenobactin thus still the intracellular iron is low. This upregulation makes the iron starvation to a manageable level.

The authors may think about this possibility.

Minor comments:

Line 35: The authors should introduce stenobactin to the readers in the abstract.

Line 82: Are all siderophores peptides?

Line 84: Citation missing for this comment

Lines 91-94: Citation missing for this section

Line 131: What is 'genomic search'? The authors may want to clarify the methods.

Line 154: The authors should mention that KJ is the name of a strain.

Line 156: The change should be 'relative', not 'absolute', as only normalized fold-changes (thus relative) have been used in this study.

Line 160: The authors should cite and mention the method of gene enrichment they used.

Line 189: What would the authors expect if they complemented the *smeYZ* strain with other efflux pumps to test redundancy?

Line 194, Fig 1B: The WT and *smeYZ* mutant have different CFUs in no treatment control. The authors should mention that in the text with clarifications or they might repeat the data with equal CFUs.

Lines 199-203 and 238: I appreciate this clarification from the authors. Although, how did they complement for Fig 2E?

Lines 205-206: The subheading could be re-written as 'Blocking stenobactin synthesis in *KJdelYZ* restores the intracellular iron levels'

Line 282: *smeE* is up not *smeD*

Line 295: If there is redundancy, why would the authors expect altered iron levels in *smeYZ* mutant?

Lines 301-303: Did the authors find any fur binding sites in the regulatory regions of *sbiAB* and *smeDEF*?

Lines 350-351: The authors should mention the figure number.

Reviewer #2 (Comments for the Author):

Roles of *SmeYZ*, *SbiAB*, and *SmeDEF* efflux systems in iron homeostasis of 1 *Stenotrophomonas maltophilia*
Wu et al

Summary

This manuscript details studies of the involvement of three different efflux pumps in siderophore export/iron homeostasis in the Gram-negative bacterium *Stenotrophomonas maltophilia*. It continues from previous observations that inactivation of a primary housekeeping pump (*SmeYZ*) of the RND family had pleiotropic effects on the cells, including loss of motility, decreased protease production and increased susceptibility to oxidative stress and antibiotics. Antibiotic susceptibility likely reflects direct efflux by *SmeXY*, and the authors demonstrate this for tobramycin. However, the other effects suggested a more complex impact of pump deletion was also occurring. Here the authors began with a transcriptomics approach comparing a pump deletion mutant to the parent strain and the profile included upregulation of many iron regulated genes and several of these genes overlapped a profile induced by treatment of cells with dipyrical implying that the cells regulatory circuits were responding to a low iron condition. This in turn suggested that *smeYZ* may be important in the secretion of the siderophore stenobactin, and, that the loss of this pump may have been increasing intracellular concentration of the siderophore causing intracellular iron sequestration. Possibly consistent with this, measurement of cellular iron levels using mass spec indicated a low level iron condition which was alleviated by deleting genes required for stenobactin synthesis. Deletion of stenobactin synthesis also reversed the pleiotropic effects of *smeYZ* deletion. Two other efflux pump genes that were also upregulated in response to deletion of *smeYZ* (*smeDEF* and *sbiAB*) were shown to be involved in stenobactin export through the use of various double and triple mutants and by testing stenobactin levels in supernatants from these cells by CAS and growth stimulation assays. From these studies a role for *smeYZ* and *smeDEF* in stenobactin export appeared clear with perhaps a smaller role for *SbiAB*. This redundancy of pump function seemed to allow for extrusion of stenobactin in the *smeYZ* mutant, suggested that stenobactin itself may not have been accumulated that much intracellularly. However, deletion of the *SbiAB* pump in a *smeYZ* background specifically reversed these effects (similar to deletion of stenobactin synthesis genes). The authors propose that *SbiAB* is somehow extruding iron and its upregulation in response to deletion of *smeYZ* is what leads to the intracellular reduced iron

stress. The authors conclude with a proposed model whereby deletion of *smeYZ* interferes with extrusion of toxic metabolites which in turn induces stress leading to stenobactin synthesis. This then induces *smeDEF* and *SbiAB* expression and one or both, extrude stenobactin, while *SbiAB* also extrudes iron. The authors propose that this iron depletion leads to the various pleiotropic effects. Some comments for consideration are listed below, in order of appearance in the manuscript.

1. Line 57; suggest being clear about your definition of multi-drug resistance. Does this mean intrinsically, or that MDR is emerging ?
2. Line 70; might be more accurate to say that the host utilizes nutritional immunity (or hypoferrremia of infection) rather than that the host cells do so.
3. Is there a Fur binding box sequence upstream of the *smeDEF* and *sbiAB* gene, and/or any regulatory proteins (e.g. two-component) whose location and expression profiles through these experiments might suggest a role in sensing stenobactin ?
4. Line 82 - what is meant by synthesized independently (of what ?)
5. Line 90 - clarify what is meant by "in the vicinity" (of what ?)
6. Line 111; what is meant by physiological niches in the context of this sentence ?
7. Line 116 -117 is incorrectly worded.
8. Suggest rewording "relatively rare" to something like "mechanisms for iron acquisition and maintaining iron homeostasis in *Sm* are not fully understood except for..." and perhaps delete line 129.
9. Line 142; the *smeYZ* pump has already been mentioned on line 138 - suggest deleting this and just stating that *SmeXY* is constitutively expressed suggesting it may be a housekeeping pump akin to *MexAB-OprM* in *P. aeruginosa* or *AcrAB-TolC* in *E. coli* (with citations).
10. Suggest delete line 147
11. Table 1 and S1 and lines 151-171;
 - a. Suggest adding the fold change and encoded protein information to Table S1 to match format of Table 1. Might want to add the DIP data to Table 1 as well so readers can compare individual genes more easily.
 - b. The emphasis here is on upregulated genes, presumably reflecting the idea that iron transport genes would be upregulated under iron starvation, but little is said regarding the downregulated genes except generally in line 164. There seems to be much less overlap of *smeXY* and DIP profiles for downregulated genes; would suggest a line or two discussing what this may mean.
 - c. The only downregulated gene included in Table 1 is the *tonB* gene, and it does not meet the 3-fold change criteria for genes considered as differentially expressed for this manuscript. Since the uptake systems are *tonB* dependent, this inclusion in Table 1 should be explained.
 - d. For Table 1, the title suggests that these genes were "selected" for transcriptomic analysis, but they are really a subset being shown in more detail from the overall RNA seq shown in Table S1 (i.e. selected "from" the transcriptome, and based on the enrichment analysis) - would reword title to more clearly link to the description of what the Table contains on line 166. Also, is the inclusion of *SmeD*, *E* and *F*, and *SbiA* and *B* here clear based on the description of Table 1 as being genes "involved in siderophore production or uptake" as these would not be in that category (were these genes included under the ontology category of "siderophore transmembrane transport GO:0044718 ? If so then should say that. Could also bold the genes for siderophore production and uptake in the Table and change the text to say the Table "includes" these (bold).
 - e. Table one lists ent genes as involved in enterobactin synthesis - would suggest clarifying somewhere that these are for stenobactin in this case
 - f. Line 168; the disturbance in homeostasis would be either an iron overload or starvation, but not a response. The response would be "to" the condition. Example "Deletion of *smeXY* appeared to cause an iron starvation condition, based on the cellular response observed here". Suggest rewording figure legend for Fig 1A as well - from "experiences" a response to "exhibits" a response.
 - g. Fig S1 is mislabelled as upregulated genes but is downregulated, again there seems to be much less correlation for the downregulated genes - is this expected ?
12. Line 181 reword to remove "simultaneously" - better to say upreg and downreg, respectively, in both. (can't be up and down at the same time)
13. One thing that isn't clear (maybe I'm not understanding) related to Figure 1B; at a concentration of 30 DIP, the KJ ent mutant is much more susceptible than the KJ *smeYZ* ent double mutant - and this is the only instance across all the tests where this difference occurs. Why would the single mutant be more susceptible than the double mutant when neither can produce stenobactin, and both mutants have essentially the same intracellular iron concentration ? Further to this - I wonder if it might be helpful use basic iron deficient medium for some experiments in this study instead of LB +/- DIP etc. How would the various

mutants in this study behave in, for example, chelex treated minimal salt medium (or even chelexed LB) where iron is really limited without any extraneous compounds ? How about using a weaker external chelator of FeIII like EDDHA ?

14. As regards the Mass spec determinations of intracellular iron levels, and related to this, the phenomenon of intracellular chelation by stenobactin: it would be worth a brief but precise description of this and what was expected. Would intracellular iron be FeII (the form that binds to DIP) - and then would this be changed upon sonication of cells and release of the iron ? And how would this affect solubility (would it all bind immediately to the accumulated stenobactin) ? You mention that stenobactin may be taking Fe from intracellular iron binding proteins (presumably as FeIII in that case) but if stenobactin is binding it inside the cells then why would intracellular iron levels be expected to decrease rather than just being moved to stenobactin ? Just suggesting a brief explanation of how sequestration would be expected to cause lower intracellular iron by mass spec.

15. Line 207 - the results showed that the genes were upregulated which suggests, but does not show, that stenobactin itself was being produced. Fig S3 - should perhaps include the smlt designation and the gene names or predicted products for all rather than a mix - requires going back to Table 1 to know what these are.

16. Line 222 - what is meant by "virtual" ? Would suggest to not use the term "side effects" in line 223.

17. It is interesting that you can complement a KJ smeYZ ent double mutant but not a KJ smeYZ single mutant - does this suggest that this is another of the pleiotropic effects of smeYZ deletion that depends on stenobactin ? Might this also be consistent with the downregulation of genes related to peptidoglycan etc and possibly suggest an impact on cell envelope integrity ? In some Gram negatives membrane damage also impacts motility. It might be worth seeing if the pump mutant is unusually susceptible to cell wall antibiotics or to compounds like rifampicin which can indicate outer membrane damage, and if so does it occur in the double knockout.

18. Figure 3C needs to indicate that this was done by RT-PCR. Also - for the various novel genes identified here that appear iron regulated - were there fur box sequences located upstream ?

19. Line 290 - the mutants themselves were not subjected to these assays but rather were used to generate cell free supernatants which were then used in the assays.

20. Following from the points made in comment 14 above, line 325 is not clear. You had already determined that intracellular iron was reduced in the smeYZ mutant, and proposed that this could affect the function of iron binding proteins, leading to the multiple effects. What you really mean here is that you were reconsidering whether this reduction could really have been mediated by a severe intracellular sequestration by stenobactin.

21. Line 336; Table S2 appears to be unnecessary. And the fold change values for SbiA and B appear to be different than they are in Table 1.

22. Figure 4B; this does suggest that the stenobactin genes are upregulated in stationary - but some mention should be made about the iron status of the culture at stationary phase. Could this just be that the culture is running low on iron which is part of why it is entering stationary phase.

23. Line 491; empty brackets

24. Would it be informative to mobilize a plasmid with sbiAB under a strong constitutive promoter into KJ and/or certain mutants to see if it lowers intracellular iron levels generally even if smeXY is present and whether or not stenobactin synthesis is disrupted ?

Staff Comments:

Preparing Revision Guidelines

- Point-by-point responses to the issues raised by the reviewers in a file named "Response to Reviewers," NOT IN YOUR COVER LETTER.
- Upload a compare copy of the manuscript (without figures) as a "Marked-Up Manuscript" file.
- Each figure must be uploaded as a separate file, and any multipanel figures must be assembled into one file.

- Manuscript: A .DOC version of the revised manuscript
- Figures: Editable, high-resolution, individual figure files are required at revision, TIFF or EPS files are preferred

Please return the manuscript within 60 days; if you cannot complete the modification within this time period, please contact me. If you do not wish to modify the manuscript and prefer to submit it to another journal, please notify me of your decision immediately so that the manuscript may be formally withdrawn from consideration by Microbiology Spectrum.

Response to reviewers

Reviewer #1 (Public repository details (Required)):

The RNAseq data should be submitted in an appropriate public database. The authors have not provided any accession number for that in the manuscript.

Reply: The accession number of RNAseq data has been provided. Please see Lines 135-136 & 565-566 in the revised manuscript.

Reviewer #1 (Comments for the Author):

*In the manuscript Spectrum02448-2, Wu et al. reported the functions of efflux pump SmeYZ of *Stenotrophomonas maltophilia* in the export of siderophore stenobactin. I found the manuscript to be very well-written. The rationale of the experiments and the analysis of the results were clearly presented. The work is interesting and will definitely contribute to the growing body of literature on the functional roles of efflux pumps in siderophore metabolism.*

There is just one thing about this work which is unclear to me:

*Lines 295, 301-303, 348-349, 365-366, 452 (Figures 1C, 3C, and 4A): Here the authors show that deletion of *smeYZ* causes low iron levels. This deletion of *smeYZ* causes upregulation of *sbiAB* and *smeDEF* which also significantly contribute to the secretion of stenobactin. How does that lead to reduction in intracellular iron levels? Wouldn't it be the opposite because upregulation of redundant efflux pumps such as *sbiAB* and *smeDEF* should reduce the toxicity of stenobactin? This is further supported by the fact that *ent* deletion restores the iron levels in *smeYZ* deletion strain. **If the authors perform an experiment where *sbiAB* and *smeDEF* are overexpressed in a *smeYZ* deletion strain, shouldn't that restore iron levels?** This is a major gap in their rationale in the mechanism of pleiotropy and thus the physiological role of *smeYZ*. The authors do not provide any clear rationale as to why iron is low in *smeYZ* strain.*

*Is the following scenario/hypothesis possible? A deletion of *smeYZ* causes stenobactin accumulation inside the cells (as the authors clearly show) causing stenobactin toxicity via iron-starvation to which the cells respond by upregulating *sbiAB* and *smeDEF* (also reported by authors). But this upregulation is not enough to release all the accumulated intracellular stenobactin thus still the intracellular iron is low. This upregulation makes the iron starvation to a manageable level.*

The authors may think about this possibility.

Reply:

(1) Suggestions to overexpress *sbiAB* and *smeDEF* in KJΔYZ

Mutant KJΔYZ has a defect in stably maintaining a plasmid (Please see Lines 176-180 in the revised manuscript). Thus, it failed to transfer the plasmids containing *sbiAB* or *smeDEF* into KJΔYZ.

(2) The possible rationale as to why iron is low in KJΔYZ

Thanks for your scenario/hypothesis. The pleiotropic defects of KJΔYZ were not observed in KJΔYZΔ*SbiAB*, except aminoglycoside susceptibility (please see Fig. 2 in the revised manuscript), which seems not to support your hypothesis. We have tried to discuss this issue. Please see Lines 274-281.

Minor comments:

Line 35: The authors should introduce stenobactin to the readers in the abstract.

Reply: The sentence has been re-written. Please see Lines 35-36 in the revised manuscript.

Line 82: Are all siderophores peptides?

Reply: The sentence has been deleted according to the editor's comment to streamline the manuscript.

Line 84: Citation missing for this comment

Reply: The reference has been cited. Please see Line 84 in the revised manuscript.

Lines 91-94: Citation missing for this section

Reply: The reference has been cited. Please see Line 90 in the revised manuscript.

Line 131: What is 'genomic search'? The authors may want to clarify the methods.

Reply: Reply: These sentences have been deleted according to the editor's comment to streamline the manuscript.

Line 154: The authors should mention that KJ is the name of a strain.

Reply: Thanks for your kind reminder. The KJΔYZ mutant has been clearly described. Please see Lines 129-130 in the revised manuscript.

Line 156: The change should be 'relative', not 'absolute', as only normalized fold-changes (thus relative) have been used in this study.

Reply: Thanks for your kind reminder. The mistake has been corrected. Please

see Line 132 in the revised manuscript.

Line 160: The authors should cite and mention the method of gene enrichment they used.

Reply: The gene enrichment method has been described in Materials and Methods and the citation has been added. Please see Line 472 in the revised manuscript.

Line 189: What would the authors expect if they complemented the smeYZ strain with other efflux pumps to test redundancy?

Reply: Due to the pleiotropic defects in smeYZ mutant, the conjugation efficiency was too low to obtain stable transconjugants when smeYZ mutant was used as a recipient strain. Thus, the plasmid-mediated complementation is not successfully performed for smeYZ mutant. This phenomenon has been described in manuscript. Please see Lines 176-180 in the revised manuscript.

Line 194, Fig 1B: The WT and smeYZ mutant have different CFUs in no treatment control. The authors should mention that in the text with clarifications or they might repeat the data with equal CFUs.

Reply: We had repeated this experience by CFU determination and results were included in Fig. S4. Please see Lines 168-171 and Fig. S4 in the revised manuscript.

Lines 199-203 and 238: I appreciate this clarification from the authors. Although, how did they complement for Fig 2E?

Reply: When KJΔYZ was used as a recipient strain, the plasmid complementation failed. We think this failure is linked to the low iron level of KJΔYZ. The recipient strains for complementation in Fig. 2E were KJΔYZΔEnt and KJΔYZΔSbiAB, whose iron levels were as high as the wild-type KJ. Thus, the plasmid pSmeYZ can be successfully transported into KJΔYZΔEnt and KJΔYZΔSbiAB by conjugation. This issue has been described and added in the Discussion section. Please see Lines 421-427 in the revised manuscript.

Lines 205-206: The subheading could be re-written as 'Blocking stenobactin synthesis in KJdelyZ restores the intracellular iron levels'

Reply: The subheading has been re-written accordingly. Please see Line 182 in the revised manuscript.

Line 282: smeE is up not smeD

Reply: Thanks for your kind reminder. The mistake has been corrected. Please see Line 238 in the revised manuscript.

Line 295: If there is redundancy, why would the authors expect altered iron levels in smeYZ mutant?

Reply: Based on our results in this study, SmeYZ, SmeDEF, and SbiAB had function redundancy in stenobactin secretion, but not in the iron levels homeostasis (please see Fig. 1 & Fig. 3 in the revised manuscript). Overexpression of SbiAB resulted in low iron levels; however, overexpression of SmeYZ or SmeDEF had no impact on the reduction of iron levels (please see Fig. 1 in the revised manuscript). This concept has been more elucidated in Discussion section. Please see Lines 428-435 in the revised manuscript.

Lines 301-303: Did the authors find any fur binding sites in the regulatory regions of sbiAB and smeDEF?

Reply: The putative Fur binding sites have been surveyed, but no positive results were obtained. Please see Lines 239-243 in the revised manuscript.

Lines 350-351: The authors should mention the figure number.

Reply: The figure numbers have been added. Please see Lines 301 & 302 in the revised manuscript.

Reviewer #2 (Comments for the Author):

*Roles of SmeYZ, SbiAB, and SmeDEF efflux systems in iron homeostasis of *Stenotrophomonas maltophilia**

Wu et al

Summary

*This manuscript details studies of the involvement of three different efflux pumps in siderophore export/iron homeostasis in the Gram-negative bacterium *Stenotrophomonas maltophilia*. It continues from previous observations that inactivation of a primary housekeeping pump (SmeYZ) of the RND family had pleiotropic effects on the cells, including loss of motility, decreased protease production and increased susceptibility to oxidative stress and antibiotics. Antibiotic susceptibility likely reflects direct efflux by SmeXY, and the authors demonstrate this*

for tobramycin. However, the other effects suggested a more complex impact of pump deletion was also occurring. Here the authors began with a transcriptomics approach comparing a pump deletion mutant to the parent strain and the profile included upregulation of many iron regulated genes and several of these genes overlapped a profile induced by treatment of cells with dipyrical implying that the cells regulatory circuits were responding to a low iron condition. This in turn suggested that smeYZ may be important in the secretion of the siderophore stenobactin, and, that the loss of this pump may have been increasing intracellular concentration of the siderophore causing intracellular iron sequestration. Possibly consistent with this, measurement of cellular iron levels using mass spec indicated a low level iron condition which was alleviated by deleting genes required for stenobactin synthesis. Deletion of stenobactin synthesis also reversed the pleiotropic effects of smeYZ deletion. Two other efflux pump genes that were also upregulated in response to deletion of smeYZ (smeDEF and sbiAB) were shown to be involved in stenobactin export through the use of various double and triple mutants and by testing stenobactin levels in supernatants from these cells by CAS and growth stimulation assays. From these studies a role for smeYZ and smeDEF in stenobactin export appeared clear with perhaps a smaller role for SbiAB. This redundancy of pump function seemed to allow for extrusion of stenobactin in the smeYZ mutant, suggested that stenobactin itself may not have been accumulated that much intracellularly. However, deletion of the SbiAB pump in a smeYZ background specifically reversed these effects (similar to deletion of stenobactin synthesis genes). The authors propose that SbiAB is somehow extruding iron and its upregulation in response to deletion of smeYZ is what leads to the intracellular reduced iron stress. The authors conclude with a proposed model whereby deletion of smeYZ interferes with extrusion of toxic metabolites which in turn induces stress leading to stenobactin synthesis. This then induces smeDEF and SbiAB expression and one or both, extrude stenobactin, while SbiAB also extrudes iron. The authors propose that this iron depletion leads to the various pleiotropic effects. Some comments for consideration are listed below, in order of appearance in the manuscript.

1. Line 57; suggest being clear about your definition of multi-drug resistance. Does this mean intrinsically, or that MDR is emerging ?

Reply: The word 'intrinsic' has been added accordingly. Please see Line 55 in the revised manuscript.

2. Line 70; might be more accurate to say that the host utilizes nutritional immunity (or hypoferremia of infection) rather than that the host cells do so.

Reply: The sentence has been re-written accordingly. Please see Lines 69-70 in the revised manuscript.

3. *Is there a Fur binding box sequence upstream of the smeDEF and sbiAB gene, and/or any regulatory proteins (e.g. two-component) whose location and expression profiles through these experiments might suggest a role in sensing stenobactin ?*

Reply: The putative Fur binding sites have been surveyed, but no positive results were obtained. Please see Lines 239-243 in the revised manuscript.

4. *Line 82 - what is meant by synthesized independently (of what ?)*

Reply: The sentence has been deleted according to the editor's comment to streamline the manuscript.

5. *Line 90 - clarify what is meant by "in the vicinity" (of what ?)*

Reply: This sentence has been re-written. Please see Line 87 in the revised manuscript.

6. *Line 111; what is meant by physiological niches in the context of this sentence?*

Reply: These sentences have been deleted according to the editor's comment to streamline the manuscript.

7. *Line 116 -117 is incorrectly worded.*

Reply: This sentence has been re-written. Please see Lines 102-104 in the revised manuscript.

8. *Suggest rewording "relatively rare" to something like "mechanisms for iron acquisition and maintaining iron homeostasis in Sm are not fully understood except for..." and perhaps delete line 129.*

Reply: These sentences have been deleted accordingly.

9. *Line 142; the smeYZ pump has already been mentioned on line 138 - suggest deleting this and just stating that SmeXY is constitutively expressed suggesting it may be a housekeeping pump akin to MexAB-OprM in P. aeruginosa or AcrAB-TolC in E. coli (with citations).*

Reply: Thanks for your suggestions. The contents have been supplemented. Please see Lines 118-119 in the revised manuscript.

10. *Suggest delete line 147*

Reply: The sentence has been deleted accordingly.

11. Table 1 and S1 and lines 151-171;

a. Suggest adding the fold change and encoded protein information to Table S1 to match format of Table 1. Might want to add the DIP data to Table 1 as well so readers can compare individual genes more easily.

Reply: Thanks for your suggestions. The Table S1 has been changed accordingly. The DIP data have been added into Table 1. Please see Table 1 and Table S1 in the revised manuscript.

b. The emphasis here is on upregulated genes, presumably reflecting the idea that iron transport genes would be upregulated under iron starvation, but little is said regarding the downregulated genes except generally in line 164. There seems to be much less overlap of *smeXY* and DIP profiles for downregulated genes; would suggest a line or two discussing what this may mean.

Reply: The discussion concerning the downregulated genes in both KJΔYZ and DIP-treatment was added. Please see Lines 376-379 in the revised manuscript.

c. The only downregulated gene included in Table 1 is the *tonB* gene, and it does not meet the 3-fold change criteria for genes considered as differentially expressed for this manuscript. Since the uptake systems are *tonB* dependent, this inclusion in Table 1 should be explained.

Reply: Thanks for your suggestion. TonB genes are generally redundant in a bacterium; thus, we resurveyed the putative TonB genes in *S. maltophilia* genome and have some interesting findings. We have added the transcriptome results of the five TonB genes into the Table 1 (Smlt0009, Smlt2939, Smlt3094 Smlt3477, and Smlt3892) and some viewpoints have been proposed in the Discussion section. Please see Table 1 and Lines 379-386 in the revised manuscript. Thank you for your carefulness again.

d. For Table 1, the title suggests that these genes were "selected" for transcriptomic analysis, but they are really a subset being shown in more detail from the overall RNA seq shown in Table S1 (i.e. selected "from" the transcriptome, and based on the enrichment analysis) - would reword title to more clearly link to the description of what the Table contains on line 166. Also, is the inclusion of *SmeD*, *E* and *F*, and *SbiA* and *B* here clear based on the description of Table 1 as being genes "involved in siderophore production or uptake" as these would not be in that category (were these genes included under the ontology category of "siderophore transmembrane transport

GO:0044718 ? If so then should say that. Could also bold the genes for siderophore production and uptake in the Table and change the text to say the Table "includes" these (bold).

Reply: The title of Table 1 has been re-written accordingly.

The genes for siderophore production and uptake in Table 1 were marked in bold. Please see Table 1 in the revised manuscript.

e. Table one lists ent genes as involved in enterobactin synthesis - would suggest clarifying somewhere that these are for stenobactin in this case

Reply: The genes involved in stenobactin synthesis have been clarified. Please see Table 1 in the revised manuscript.

f. Line 168; the disturbance in homeostasis would be either an iron overload or starvation, but not a response. The response would be "to" the condition. Example "Deletion of smeXY appeared to cause an iron starvation condition, based on the cellular response observed here". Suggest rewording figure legend for Fig 1A as well - from "experiences" a response to "exhibits" a response.

Reply: The words have been reworded accordingly. Please see Line 143 and the figure legend of Fig. 1 (Line 718) in the revised manuscript.

g. Fig S1 is mislabelled as upregulated genes but is downregulated, again there seems to be much less correlation for the downregulated genes - is this expected ?

Reply: The mistakes have been corrected accordingly. The discussion concerning the downregulated genes in both KJ Δ YZ and DIP-treatment was added. Please see Lines 376-379 and the figure legend of Fig. S1 in the revised manuscript.

12. Line 181 reword to remove "simultaneously" - better to say upreg and downreg, respectively, in both. (can't be up and down at the same time)

Reply: This sentence has been corrected accordingly. Please see Line 155 in the revised manuscript.

13. One thing that isn't clear (maybe I'm not understanding) related to Figure 1B; at a concentration of 30 DIP, the KJ ent mutant is much more susceptible than the KJ smeYZ ent double mutant - and this is the only instance across all the tests where this difference occurs. Why would the single mutant be more susceptible than the double mutant when neither can produce stenobactin, and both mutants have essentially the same intracellular iron concentration ? Further to this - I wonder if it might be helpful use basic iron deficient medium for some experiments in this study instead of

LB +/- DIP etc. How would the various mutants in this study behave in, for example, chelex treated minimal salt medium (or even chelexed LB) where iron is really limited without any extraneous compounds ? How about using a weaker external chelator of FeIII like EDDHA ?

Reply: Thanks for your kind reminder. We rechecked our raw data and repeated this experiment. The picture for KJΔYZΔEnt at the concentration of 30 DIP was a misprint in the original manuscript. This mistake has been corrected. Please see Fig. 1. in the revised manuscript. Thank you for your carefulness again.

14. As regards the Mass spec determinations of intracellular iron levels, and related to this, the phenomenon of intracellular chelation by stenobactin: it would be worth a brief but precise description of this and what was expected. Would intracellular iron be FeII (the form that binds to DIP) - and then would this be changed upon sonication of cells and release of the iron ? And how would this affect solubility (would it all bind immediately to the accumulated stenobactin) ? You mention that stenobactin may be taking Fe from intracellular iron binding proteins (presumably as FeIII in that case) but if stenobactin is binding it inside the cells then why would intracellular iron levels be expected to decrease rather than just being moved to stenobactin ? Just suggesting a brief explanation of how sequestration would be expected to cause lower intracellular iron by mass spec.

Reply: I totally agreed your viewpoints. I am sorry that the writing in the first manuscript is not clear enough. The viewpoint has been further clearly elucidated in the revised manuscript.

Please see Lines 274-281 in the revised manuscript.

15. Line 207 - the results showed that the genes were upregulated which suggests, but does not show, that stenobactin itself was being produced. Fig S3 - should perhaps include the smlt designation and the gene names or predicted products for all rather than a mix - requires going back to Table 1 to know what these are.

Reply: The word “showed” has been replaced with “suggested” accordingly. Please see Line 184 in the revised manuscript.

The Smlt designation has been added to Fig. S3. Please see Fig. S3 in the revised manuscript.

16. Line 222 - what is meant by "virtual" ? Would suggest to not use the term "side effects" in line 223.

Reply: These sentences have been deleted according to the editor’s comment to

streamline the manuscript.

17. It is interesting that you can complement a KJ smeYZ ent double mutant but not a KJ smeYZ single mutant - does this suggest that this is another of the pleiotropic effects of smeYZ deletion that depends on stenobactin? Might this also be consistent with the downregulation of genes related to peptidoglycan etc and possibly suggest an impact on cell envelope integrity? In some Gram negatives membrane damage also impacts motility. It might be worth seeing if the pump mutant is unusually susceptible to cell wall antibiotics or to compounds like rifampicin which can indicate outer membrane damage, and if so does it occur in the double knockout.

Reply: When KJΔYZ was used as a recipient strain, the plasmid complementation failed. We think this failure is linked to the low iron level of KJΔYZ. The iron levels of KJΔYZΔEnt and KJΔYZΔSbiAB were as high as the wild-type KJ. Thus, the plasmid pSmeYZ can be successfully transported into KJΔYZΔEnt and KJΔYZΔSbiAB by conjugation. This issue has been described and added in the Discussion section. Please see Lines 421-427 in the revised manuscript.

In addition, the antibiotic susceptibility of KJ and KJΔYZ have been tested in our published paper (Antimicrob. Agents Chemother. 2015. 59:4067-4073). The β-lactam susceptibilities of KJ and KJΔYZ are comparable.

18. Figure 3C needs to indicate that this was done by RT-PCR. Also - for the various novel genes identified here that appear iron regulated - were there fur box sequences located upstream ?

Reply: The word “qRT-PCR” has been added into the figure legend of Fig. 3. Please see the figure legend of Fig. 3 (Line 782) in the revised manuscript.

The “Fur box” analysis has added. Please see Lines 239-243 in the revised manuscript.

19. Line 290 - the mutants themselves were not subjected to these assays but rather were used to generate cell free supernatants which were then used in the assays.

This sentence has been re-written. Please see Lines 249-250 in the revised manuscript.

20. Following from the points made in comment 14 above, line 325 is not clear. You had already determined that intracellular iron was reduced in the smeYZ mutant, and proposed that this could affect the function of iron binding proteins, leading to the multiple effects. What you really mean here is that you were reconsidering whether

this reduction could really have been mediated by a severe intracellular sequestration by stenobactin.

Reply: I am sorry that the writing in the first manuscript is not clear enough. The paragraph has been re-written in the revised manuscript. Please see Lines 274-281 in the revised manuscript.

21. Line 336; Table S2 appears to be unnecessary. And the fold change values for SbiA and B appear to be different than they are in Table 1.

Reply: Table S2 has been deleted accordingly. The Table 1 has been carefully rechecked.

22. Figure 4B; this does suggest that the stenobactin genes are upregulated in stationary - but some mention should be made about the iron status of the culture at stationary phase. Could this just be that the culture is running low on iron which is part of why it is entering stationary phase.

Reply: The intracellular iron level of stationary-phase KJ cells has been determined by ICP-MS. Please see Lines 338, 340, and 344-347 and Fig. 1C in the revised manuscript.

23. Line 491; empty brackets

Reply: The empty bracekets have been deleted.

24. Would it be informative to mobilize a plasmid with sbiAB under a strong constitutive promoter into KJ and/or certain mutants to see if it lowers intracellular iron levels generally even if smeXY is present and whether or not stenobactin synthesis is disrupted ?

Reply: The experiments have been added. Please see Lines 309-318 and Fig. S5 in the revised manuscript.

March 6, 2022

Prof. Tsuey-Ching Yang
National Yang Ming Chiao Tung University
155 Section 2, Lie-Nong Street,
Taipei, Taiwan 112
Taiwan

Re: Spectrum02448-21R1 (Roles of SmeYZ, SbiAB, and SmeDEF efflux systems in iron homeostasis of *Stenotrophomonas maltophilia*.)

Dear Prof. Tsuey-Ching Yang:

Thank you for submitting your manuscript to Microbiology Spectrum. Your manuscript was reviewed by the same two reviewers who reviewed the initial submission. You will note that one of the reviewers is still not fully satisfied by the revisions you made. The reviewer has made some critical suggestions that need to be addressed before I can make a decision on your manuscript. When submitting the revised version of your paper, please provide (1) point-by-point responses to the issues raised by the reviewers as file type "Response to Reviewers," not in your cover letter, and (2) a PDF file that indicates the changes from the original submission (by highlighting or underlining the changes) as file type "Marked Up Manuscript - For Review Only". Please use this link to submit your revised manuscript - we strongly recommend that you submit your paper within the next 60 days or reach out to me. Detailed instructions on submitting your revised paper are below.

Link Not Available

Sincerely,

Ayush Kumar

Journals Department
Reviewer comments:

Reviewer #1 (Comments for the Author):

The authors have sufficiently addressed my concerns.

Reviewer #2 (Comments for the Author):

see attached file

Staff Comments:

Preparing Revision Guidelines

Please return the manuscript within 60 days; if you cannot complete the modification within this time period, please contact me. If you do not wish to modify the manuscript and prefer to submit it to another journal, please notify me of your decision immediately so that the manuscript may be formally withdrawn from consideration by Microbiology Spectrum.

Spectrum02448-21R1

Summary

Several modifications have been made that have improved the manuscript. I have listed a few additional comments.

Figures cited in discussion makes the discussion seem a bit like results in places.

Line 70; might be more accurate to say that the host utilizes nutritional immunity (or hypoferrremia of infection) rather than that the host cells do so.

Still says that the host “cells” do this.

Is there a Fur binding box sequence upstream of the smeDEF and sbiAB gene, and/or any regulatory proteins (e.g. two-component) whose location and expression profiles through these experiments might suggest a role in sensing stenobactin ? **Reply: The putative Fur binding sites have been surveyed, but no positive results were obtained. Please see Lines 239-243 in the revised manuscript.**

OK but what inference do you draw from this ? and what does it mean that you “tried” to survey for them ?

Line 116 -117 is incorrectly worded. **Reply: This sentence has been re-written. Please see Lines 102-104 in the revised manuscript.**

Still not worded optimally. Do you mean here that S maltophilia is generally an environmental organism but is an opportunistic pathogen that can infect a broad range of hosts under the right conditions ? You refer to Sm as opportunistic elsewhere in the document.

Line 142; the smeYZ pump has already been mentioned on line 138 -suggest deleting this and just stating that SmeXY is constitutively expressed suggesting it may be a housekeeping pump akin to MexAB-OprM in P. aeruginosa or AcrAB-TolC in E. coli (with citations). **Reply: Thanks for your suggestions. The contents have been supplemented. Please see Lines 118-119 in the revised manuscript.**

Still unclear – perhaps delete “Of these efflux pumps, the SmeYZ is worth mentioning”; More importantly though – the suggestion wasn’t to compare the expression levels to other housekeeping pumps, but to indicate simply that YZ appears to be constitutive, and that is a characteristic of housekeeping pumps. Would avoid getting into what is “highly” expressed.

emphasis here is on upregulated genes, presumably reflecting the idea that iron transport genes would be upregulated under iron starvation, but little is said regarding the downregulated genes except generally in line 164. There seems to be much less overlap of smeXY and DIP profiles for downregulated genes; would suggest a line or two discussing what this may mean. **Reply: The discussion concerning the downregulated genes in both KJ□YZ and DIP-treatment was added. Please see Lines 376-379 in the revised manuscript**

It was added to text that differences were observed, but still lacks any hypothesis about why this was the case and context. Because you state that DIP profile should be the same if your hypothesis is correct- (see line 143) then difference may become relevant. If it is that both profiles should simply

capture iron regulated genes within their respective profiles, then that would just suggest DIP treatment was mainly as a control for that subset of genes as opposed to implications about the importance of there being similarity of the expected overall profiles. Indeed, lines 139-143 indicate that sufficient analysis was possible in the absence of any DIP related profiling to conclude that stenobactin and other iron regulated genes were upregulated.

The only downregulated gene included in Table 1 is the *tonB* gene, and it does not meet the 3-fold change criteria for genes considered as differentially expressed for this manuscript. Since the uptake systems are *tonB* dependent, this inclusion in Table 1 should be explained **Reply: Thanks for your suggestion. TonB genes are generally redundant in a bacterium; thus, we resurveyed the putative TonB genes in *S. maltophilia* genome and have some interesting findings. We have added the transcriptome results of the five TonB genes into the Table 1 (Smlt0009, Smlt2939, Smlt3094 Smlt3477, and Smlt3892) and some viewpoints have been proposed in the Discussion section. Please see Table 1 and Lines 379-386 in the revised manuscript. Thank you for your carefulness again**

OK but now you need to explain what you mean by “bacterial kinetic energy” and, also the way sentence 385-386 is worded appears incorrect. It implies that YZ and DIP treatment cause the same (albeit “distinct”) impact. (Perhaps something like “Therefore *smeYZ* deletion and DIP treatment appear to cause distinct (or different) impacts on bacterial...); but this again makes it important to say something about why these *tonB* patterns may differ, given your contention that this comparison with DIP treatment is to be assessed to some extent based on how similar it is to the YZ knockout. Alternatively, suggest why it doesn't affect any conclusions from this study (and this relates back to how you presented the reason for doing the DIP-treatment profiling

As regards the Mass spec determinations of intracellular iron levels, and related to this, the phenomenon of intracellular chelation by stenobactin: it would be worth a brief but precise description of this and what was expected. Would intracellular iron be FeII (the form that binds to DIP) -and then would this be changed upon sonication of cells and release of the iron ? And how would this affect solubility (would it all bind immediately to the accumulated stenobactin) ? You mention that stenobactin may be taking Fe from intracellular iron binding proteins (presumably as FeIII in that case) but if stenobactin is binding it inside the cells then why would intracellular iron levels be expected to decrease rather than just being moved to stenobactin ? Just suggesting a brief explanation of how sequestration would be expected to cause lower intracellular iron by mass spec. **Reply: I totally agreed your viewpoints. I am sorry that the writing in the first manuscript is not clear enough. The viewpoint has been further clearly elucidated in the revised manuscript. Please see Lines 274-281 in the revised manuscript.**

The concept of what is being measured was not addressed but since the theory of sequestration seems to have been deprioritized it may not be that important. However, the edits at lines 274-281 state that sequestration cannot cause the iron level changes, therefore sequestration per se is not likely to be the mechanism of pleiotropic effects. But, on line 172-175 (earlier in the document) you state that the intracellular iron was significantly lower in KJYZ than wild-type. Knowing that the iron was lower in the mutant, and that all indications were that there could be upregulated stenobactin, you hypothesize on line 188-189 that stenobactin accumulation could chelate iron from iron binding proteins. Would check these sections and try to align the thoughts overall (or perhaps I'm misunderstanding something)

As well:

Point i isn't clear. Even if it can still export enough stenobactin for growth support, this doesn't mean intracellular levels aren't also increased if export is sub optimal

Point ii. This is simply a statement but not explained why "it cannot" cause an impact on levels; and perhaps should be said it is unlikely to explain the significant drop seen in Fig 1 – as written this implies that Fig 1 itself proves that steno accum could not do this but what I think you mean is that sequestration is unlikely to explain the difference shown in Fig 1.

It is interesting that you can complement a KJ smeYZ ent double mutant but not a KJ smeYZ single mutant -does this suggest that this is another of the pleiotropic effects of smeYZ deletion that depends on stenobactin? Might this also be consistent with the downregulation of genes related to peptidoglycan etc and possibly suggest an impact on cell envelope integrity? In some Gram negatives membrane damage also impacts motility. It might be worth seeing if the pump mutant is unusually susceptible to cell wall antibiotics or to compounds like rifampicin which can indicate outer membrane damage, and if so does it occur in the double knockout.
Reply: When KJ□YZ was used as a recipient strain, the plasmid complementation failed. We think this failure is linked to the low iron level of KJ□YZ. The iron levels of KJ□YZ□Ent and KJ□YZ□SbiAB were as high as the wild-type KJ. Thus, the plasmid pSmeYZ can be successfully transported into KJ□YZ□Ent and KJ□YZ□SbiAB by conjugation. This issue has been described and added in the Discussion section. Please see Lines 421-427 in the revised manuscript.

OK but might be a little strong to say "highly" rather than "suggests". And as a general rule throughout the manuscript there are instances where you might say that something like iron levels or other things like stenobactin may "directly or indirectly" influence something else

Following from the points made in comment 14 above, line 325 is not clear. You had already determined that intracellular iron was reduced in the smeYZ mutant, and proposed that this could affect the function of iron binding proteins, leading to the multiple effects. What you really mean here is that you were reconsidering whether this reduction could really have been mediated by a severe intracellular sequestration by stenobactin

Reply: I am sorry that the writing in the first manuscript is not clear enough. The paragraph has been re-written in the revised manuscript. Please see Lines 274-281 in the revised manuscript

See above comments regarding lines 274-281

Figure 4B; this does suggest that the stenobactin genes are upregulated in stationary -but some mention should be made about the iron status of the culture at stationary phase. Could this just be that the culture is running low on iron which is part of why it is entering stationary phase

Reply: The intracellular iron level of stationary-phase KJ cells has been determined by ICP-MS. Please see Lines 338, 340, and 344-347 and Fig. 1C in the revised manuscript

Perhaps I wasn't clear. I was referring to the culture (medium) becoming progressively iron depleted. This would then be counteracted by progressively upregulating stenobactin to obtain iron. Therefore, stenobactin might be induced simply by progressive iron depletion in the culture, but its production occurs to maintain iron uptake and for some time at least if it was providing iron then intracellular levels might not change much. What evidence is there that the upregulated stenobactin production seen here is caused by toxic intermediates/metabolites vs simply depletion of iron from the medium? Is the medium iron rich at stationary?

Several modifications have been made that have improved the manuscript. I have listed a few additional comments.

Figures cited in discussion makes the discussion seem a bit like results in places.

Line 70; might be more accurate to say that the host utilizes nutritional immunity (or hypoferremia of infection) rather than that the host cells do so.

Still says that the host “cells” do this.

Reply (2nd revision): The sentence has been corrected accordingly. The word “cells” has been deleted. Please see Lines 66-67 in the revised manuscript.

Is there a Fur binding box sequence upstream of the *smeDEF* and *sbiAB* gene, and/or any regulatory proteins (e.g. two-component) whose location and expression profiles through these experiments might suggest a role in sensing stenobactin ?

Reply (1st revision): The putative Fur binding sites have been surveyed, but no positive results were obtained. Please see Lines 239-243 in the revised manuscript.

OK but what inference do you draw from this? and what does it mean that you “tried” to survey for them?

Reply (2nd round): The sentence has been rewritten and the inference has been included. Please see Lines 228-231 in the revised manuscript.

Line 116 -117 is incorrectly worded.

Reply (1st revision): This sentence has been re-written. Please see Lines 102-104 in the revised manuscript.

Still not worded optimally. Do you mean here that *S. maltophilia* is generally an environmental organism but is an opportunistic pathogen that can infect a broad range of hosts under the right conditions ? You refer to *Sm* as opportunistic elsewhere in the document.

Reply (2nd revision): The sentence has been corrected accordingly. Please see Lines 97-98 in the revised manuscript.

Line 142; the *smeYZ* pump has already been mentioned on line 138 -suggest deleting this and just stating that *SmeXY* is constitutively expressed suggesting it may be a housekeeping pump akin to *MexAB-OprM* in *P. aeruginosa* or *AcrAB-TolC* in *E. coli* (with citations).

Reply (1st revision): Thanks for your suggestions. The contents have been supplemented. Please see Lines 118-119 in the revised manuscript.

Still unclear – perhaps delete “Of these efflux pumps, the *SmeYZ* is worth mentioning”; More importantly though – the suggestion wasn’t to compare the expression levels to other housekeeping pumps, but to indicate simply that *YZ* appears to be constitutive, and that is a characteristic of housekeeping pumps. Would avoid getting into what is “highly” expressed.

Reply (2nd revision): Thanks for your suggestions. The sentences have been corrected accordingly. Please see Lines 111-113 in the revised manuscript.

emphasis here is on upregulated genes, presumably reflecting the idea that iron transport genes would be upregulated under iron starvation, but little is said regarding the downregulated genes except generally in line 164. There seems to be much less overlap of smeXY and DIP profiles for downregulated genes; would suggest a line or two discussing what this may mean.

Reply (1st revision): The discussion concerning the downregulated genes in both KJΔYZ and DIP-treatment was added. Please see Lines 376-379 in the revised manuscript

It was added to text that differences were observed, but still lacks any hypothesis about why this was the case and context. Because you state that DIP profile should be the same if your hypothesis is correct- (see line 143) then difference may become relevant. If it is that both profiles should simply capture iron regulated genes within their respective profiles, then that would just suggest DIP treatment was mainly as a control for that subset of genes as opposed to implications about the importance of there being similarity of the expected overall profiles. Indeed, lines 139-143 indicate that sufficient analysis was possible in the absence of any DIP related profiling to conclude that stenobactin and other iron regulated genes were upregulated.

Reply (2nd revision): Thanks for your suggestions. The sentences have been rewritten. Please see Lines 371-376 in the revised manuscript.

The only downregulated gene included in Table 1 is the tonB gene, and it does not meet the 3-fold change criteria for genes considered as differentially expressed for this manuscript. Since the uptake systems are tonB dependent, this inclusion in Table 1 should be explained

Reply (1st revision): Thanks for your suggestion. TonB genes are generally redundant in a bacterium; thus, we resurveyed the putative TonB genes in *S. maltophilia* genome and have some interesting findings. We have added the transcriptome results of the five TonB genes into the Table 1 (Smlt0009, Smlt2939, Smlt3094 Smlt3477, and Smlt3892) and some viewpoints have been proposed in the Discussion section. Please see Table 1 and Lines 379-386 in the revised manuscript. Thank you for your carefulness again

OK but now you need to explain what you mean by “bacterial kinetic energy” and, also the way sentence 385-386 is worded appears incorrect. It implies that YZ and DIP treatment cause the same (albeit “distinct”) impact. (Perhaps something like “Therefore smeYZ deletion and DIP treatment appear to cause distinct (or different) impacts on bacterial...); but this again makes it important to say something about why these tonB patterns may differ, given your contention that this comparison with DIP treatment is to be assessed to some extent based on how similar it is to the YZ knockout. Alternatively, suggest why it doesn’t affect any conclusions from this study (and this relates back to how you presented the reason for doing the DIP-treatment profiling

Reply (2nd revision): The sentences have been rewritten. Please see Lines 376-391 in the revised manuscript.

As regards the Mass spec determinations of intracellular iron levels, and related to this, the phenomenon of intracellular chelation by stenobactin: it would be worth a brief but precise

description of this and what was expected. Would intracellular iron be FeII (the form that binds to DIP) -and then would this be changed upon sonication of cells and release of the iron ? And how would this affect solubility (would it all bind immediately to the accumulated stenobactin) ? You mention that stenobactin may be taking Fe from intracellular iron binding proteins (presumably as FeIII in that case) but if stenobactin is binding it inside the cells then why would intracellular iron levels be expected to decrease rather than just being moved to stenobactin ? Just suggesting a brief explanation of how sequestration wouldbe expected to cause lower intracellular iron by mass spec.

Reply (1st revision): I totally agreed your viewpoints. I am sorry that the writing in the first manuscript is not clear enough. The viewpoint has been further clearly elucidated in the revised manuscript.

Please see Lines 274-281 in the revised manuscript.

The concept of what is being measured was not addressed but since the theory of sequestration seems to have been deprioritized it may not be that important. However, the edits at lines 274-281 state that sequestration cannot cause the iron level changes, therefore sequestration per se is not likely to be the mechanism of pleiotropic effects. But, on line 172-175 (earlier in the document) you state that the intracellular iron was significantly lower in KJYZ than wild-type. Knowing that the iron was lower in the mutant, and that all indications were that there could be upregulated stenobactin, you hypothesize on line 188-189 that stenobactin accumulation could chelate iron from iron binding proteins. Would check these sections and try to align the thoughts overall (or perhaps I'm misunderstanding something)

As well:

Point i isn't clear. Even if it can still export enough stenobactin for growth support, this doesn't mean intracellular levels aren't also increased if export is sub optimal

Poinjt ii. This is simply a statement but not explained why "it cannot" cause an impact on levels; and perhaps should be said it is unlikely to explain the significant drop seen in Fig 1 – as written this implies that Fig 1 itself proves that steno accum could not do this but what I think you mean is that sequestration is unlikely to explain the difference shown in Fig 1.

Reply (2nd revision): The paragraph has been rewritten. Please see Lines 264-276 in the revised manuscript.

It is interesting that you can complement a KJ smeYZ ent double mutant but not a KJ smeYZ single mutant -does this suggest that this is another of the pleiotropic effects of smeYZ deletion that depends on stenobactin? Might this also be consistent with the downregulation of genes related to peptidoglycan etc and possibly suggest an impact on cell envelope integrity? In some Gram negatives membrane damage also impacts motility. It might be worth seeing if the pump mutant is unusually susceptible to cell wall antibiotics or to compounds like rifampicin which can indicate outer membrane damage, and if so does it occur in the double knockout.

Reply (1st revision): When KJYZ was used as a recipient strain, the plasmid complementation failed. We think this failure is linked to the low iron level of KJΔYZ. The iron levels of KJΔYZΔEnt and KJΔYZΔSbiAB were as high as the wild-type KJ. Thus, the plasmid pSmeYZ can be successfully

transported into KJΔYZΔEnt and KJΔYZΔSbiAB by conjugation. This issue has been described and added in the Discussion section. Please see Lines 421-427 in the revised manuscript.

OK but might be a little strong to say “highly” rather than “suggests”. And as a general rule throughout the manuscript there are instances where you might say that something like iron levels or other things like stenobactin may “directly or indirectly” influence something else

Reply (2nd revision): The sentences have been amended. Please see Line 429 in the revised manuscript.

Following from the points made in comment 14 above, line 325 is not clear. You had already determined that intracellular iron was reduced in the smeYZ mutant, and proposed that this could affect the function of iron binding proteins, leading to the multiple effects. What you really mean here is that you were reconsidering whether this reduction could really have been mediated by a severe intracellular sequestration by stenobactin

Reply (1st revision): I am sorry that the writing in the first manuscript is not clear enough. The paragraph has been re-written in the revised manuscript. Please see Lines 274-281 in the revised manuscript

See above comments regarding lines 274-281

Reply (2nd revision): The paragraph has been rewritten. Please see Lines 264-276 in the revised manuscript.

Figure 4B; this does suggest that the stenobactin genes are upregulated in stationary -but some mention should be made about the iron status of the culture at stationary phase. Could this just be that the culture is running low on iron which is part of why it is entering stationary phase

Reply (1st revision): The intracellular iron level of stationary-phase KJ cells has been determined by ICP-MS. Please see Lines 338, 340, and 344-347 and Fig. 1C in the revised manuscript

Perhaps I wasn't clear. I was referring to the culture (medium) becoming progressively iron depleted. This would then be counteracted by progressively upregulating stenobactin to obtain iron. Therefore, stenobactin might be induced simply by progressive iron depletion in the culture, but its production occurs to maintain iron uptake and for some time at least if it was providing iron then intracellular levels might not change much. What evidence is there that the upregulated stenobactin production seen here is caused by toxic intermediates/metabolites vs simply depletion of iron from the medium ? Is the medium iron rich at stationary?

Reply (2nd revision): To my knowledge, the stimulus to upregulate stenobactin synthesis is mainly the low intracellular iron level, less related to medium iron level. However, thanks for your comments. Integrating your comments, we have rewritten the sentences. Please see Lines 338-342 in the revised manuscript.

April 17, 2022

Prof. Tsuey-Ching Yang
National Yang Ming Chiao Tung University
155 Section 2, Lie-Nong Street,
Taipei, Taiwan 112
Taiwan

Re: Spectrum02448-21R2 (Roles of SmeYZ, SbiAB, and SmeDEF efflux systems in iron homeostasis of *Stenotrophomonas maltophilia*.)

Dear Prof. Tsuey-Ching Yang:

Thank you for submitting the revised version of your manuscript. Can you please make sure that the marked version of the manuscript accurately highlights the changes you have made. There are a number of changes indicated in your response letter that are not highlighted in the 'marked up' version of the manuscript. Accurately marked changes will facilitate a quick decision. Detailed instructions on submitting your revised paper are below.

Link Not Available

Sincerely,

Ayush Kumar

Journals Department
Staff Comments:

Preparing Revision Guidelines

- Point-by-point responses to the issues raised by the reviewers in a file named "Response to Reviewers," NOT IN YOUR COVER LETTER.

- Upload a compare copy of the manuscript (without figures) as a "Marked-Up Manuscript" file.
- Each figure must be uploaded as a separate file, and any multipanel figures must be assembled into one file.
- Manuscript: A .DOC version of the revised manuscript
- Figures: Editable, high-resolution, individual figure files are required at revision, TIFF or EPS files are preferred

Please return the manuscript within 60 days; if you cannot complete the modification within this time period, please contact me. If you do not wish to modify the manuscript and prefer to submit it to another journal, please notify me of your decision immediately so that the manuscript may be formally withdrawn from consideration by Microbiology Spectrum.

Response to reviewer

Still says that the host “cells” do this.

Reply (2nd revision): The sentence has been corrected accordingly. The word “cells” has been deleted. Please see Lines 66-67 in the revised manuscript.

OK but what if any inference do you draw from this? and what does it mean that you “tried” to survey for them?

Reply (2nd round): The sentence has been rewritten and the inference has been included. Please see Lines 234-237 in the revised manuscript.

*Still not worded optimally. Do you mean here that *S maltophilia* is generally an environmental organism but is an opportunistic pathogen that can infect a broad range of hosts under the right conditions ? You refer to *Sm* as opportunistic elsewhere in the document.*

Reply (2nd revision): The sentence has been corrected accordingly. Please see Lines 99-100 in the revised manuscript.

*Still unclear – perhaps delete “Of these efflux pumps, the *SmeYZ* is worth mentioning”; More importantly though – the suggestion wasn’t to compare the expression levels to other housekeeping pumps, but to indicate simply that *YZ* appears to be constitutive, and that is a characteristic of housekeeping pumps. Would avoid getting into what is “highly” expressed.*

Reply (2nd revision): Thanks for your suggestions. The sentences have been corrected accordingly. Please see Lines 113-116 in the revised manuscript.

*It was added to text that differences were observed, but still lacks any hypothesis about why this was the case and context. Because you state that *DIP* profile should be the same if your hypothesis is correct- (see line 143) then difference may become relevant. If it is that both profiles should simply capture iron regulated genes within their respective profiles, then that would just suggest *DIP* treatment was mainly as a control for that subset of genes as opposed to implications about the importance of there being similarity of the expected overall profiles. Indeed, lines 139-143 indicate that sufficient analysis was possible in the absence of any *DIP* related profiling to conclude that *stenobactin* and other iron regulated genes were upregulated.*

Reply (2nd revision): Thanks for your suggestions. The sentences have been rewritten. Please see Lines 379-384 in the revised manuscript.

*OK but now you need to explain what you mean by “bacterial kinetic energy” and, also the way sentence 385-386 is worded appears incorrect. It implies that *YZ* and *DIP* treatment cause the same (albeit “distinct”) impact. (Perhaps something like “Therefore *smeYZ* deletion and *DIP* treatment appear to cause distinct (or different) impacts on bacterial...); but this again makes it important to say something about why these *tonB* patterns may differ, given your contention that this comparison*

with DIP treatment is to be assessed to some extent based on how similar it is to the YZ knockout. Alternatively, suggest why it doesn't affect any conclusions from this study (and this relates back to how you presented the reason for doing the DIP-treatment profiling

Reply (2nd revision): The sentences have been rewritten. Please see Lines 384-399 in the revised manuscript.

The concept of what is being measured was not addressed but since the theory of sequestration seems to have been deprioritized it may not be that important. However, the edits at lines 274-281 state that sequestration cannot cause the iron level changes, therefore sequestration per se is not likely to be the mechanism of pleiotropic effects. But, on line 172-175 (earlier in the document) you state that the intracellular iron was significantly lower in KJYZ than wild-type. Knowing that the iron was lower in the mutant, and that all indications were that there could be upregulated stenobactin, you hypothesize on line 188-189 that stenobactin accumulation could chelate iron from iron binding proteins. Would check these sections and try to align the thoughts overall (or perhaps I'm misunderstanding something)

As well:

Point i isn't clear. Even if it can still export enough stenobactin for growth support, this doesn't mean intracellular levels aren't also increased if export is sub optimal

Poinjt ii. This is simply a statement but not explained why "it cannot" cause an impact on levels;and perhaps should be said it is unlikely to explain the significant drop seen in Fig 1 – as written this implies that Fig 1 itself proves that steno accum could not do this but what I think you mean is that sequestration is unlikely to explain the difference shown in Fig 1.

Reply (2nd revision): The paragraph has been rewritten. Please see Lines 270-283 in the revised manuscript.

OK but might be a little strong to say "highly" rather than "suggests". And as a general rule throughout the manuscript there are instances where you might say that something like iron levels or other things like stenobactin may "directly or indirectly" influence something else

Reply (2nd revision): The sentences have been amended. Please see Line 438 in the revised manuscript.

See above comments regarding lines 274-281

Reply (2nd revision): The paragraph has been rewritten. Please see Lines 270-283 in the revised manuscript.

Perhaps I wasn't clear. I was referring to the culture (medium) becoming progressively iron depleted. This would then be counteracted by progressively upregulating stenobactin to obtain iron. Therefore, stenobactin might be induced simply by progressive iron depletion in the culture, but its production occurs to maintain iron uptake and for some time at least if it was providing iron then intracellular levels might not change much. What evidence is there that the upregulated stenobactin

production seen here is caused by toxic intermediates/metabolites vs simply depletion of iron from the medium ? Is the medium iron rich at stationary?

Reply (2nd revision): To my knowledge, the stimulus to upregulate stenobactin synthesis is mainly the low intracellular iron level, less related to medium iron level. However, thanks for your comments. Integrating your comments, we have rewritten the sentences. Please see Lines 346-350 in the revised manuscript.

April 30, 2022

Prof. Tsuey-Ching Yang
National Yang Ming Chiao Tung University
155 Section 2, Lie-Nong Street,
Taipei, Taiwan 112
Taiwan

Re: Spectrum02448-21R3 (Roles of SmeYZ, SbiAB, and SmeDEF efflux systems in iron homeostasis of *Stenotrophomonas maltophilia*.)

Dear Prof. Tsuey-Ching Yang:

Your manuscript has been accepted, and I am forwarding it to the ASM Journals Department for publication. You will be notified when your proofs are ready to be viewed.

Sincerely,

Ayush Kumar
Editor, Microbiology Spectrum
